# Surface ozone impacts on major crop production in China from 2010 to 2017

Dianyi Li[1], Drew Shindell[1,2], Dian Ding[3], Xiao Lu[4,5a], Lin Zhang[4], Yuqiang Zhang[1]*

[1]Nicholas School of the Environment, Duke University, 9 Circuit Dr, Durham, NC 27708

[2]Porter School of the Environment and Earth Sciences, Tel Aviv University, Tel Aviv, Israel

[3]State Key Joint Laboratory of Environmental Simulation and Pollution Control, School of Environment, Tsinghua University, Beijing, China

[4]Laboratory for Climate and Ocean-Atmosphere Studies, Department of Atmospheric and Oceanic Sciences, School of Physics, Peking University, Beijing 100871, China

[5]John A. Paulson School of Engineering and Applied Sciences, Harvard University, Cambridge, Massachusetts 02138, United Statess

[a]Now at School of Atmospheric Sciences, Sun Yat-sen University, Zhuhai, Guangdong 519082, People's Republic of Chinas

*Correspondence to*: Yuqiang Zhang (Yuqiang.Zhang@duke.edu; YuqiangZhang.thu@gmail.com)

**Abstract.** Exposure to elevated surface ozone is damaging to crops. In this study, we performed an analysis of temporal and

spatial distributions of relative yield losses (RYLs) attributable to surface ozone for major crops in China from 2010 to 2017, by applying the AOT40 metrics (hourly ozone concentration over a threshold of 40 ppbv during the growing season) simulated by chemical transport model. The major crops in China include wheat, rice (including double early & late rice, and single rice), maize (including north and south maize) and soybean. The aggregated production and associated economic losses in China and major provinces were evaluated by combing annual crop production yields and crop market prices. We estimated that, the

national annual average AOT40 in China increased from 21.98 ppm h in 2010 to 23.85 ppm h in 2017, with a peak value of 35.69 ppm h in 2014, as simulated from the model. There exists significant spatial heterogeneity for the AOT40 and RYLs across the four crops due to the seasonal ozone variations. We calculated that national mean RYLs for wheat, rice, maize and soybean were 11.45%−19.74%, 7.59%−9.29%, 0.07%−3.35%, and 6.51%−9.92% individually from 2010 to 2017. The associated crop yield losses were estimated 13.81–36.51 million metric tons (Mt), 16.89–20.03 Mt, 4.59–8.17 Mt, and 1.09–

1.84 Mt respectively, which accounted for annual average economic loss of USD 9.55 billion, USD8.53 billion, USD2.23 billion, and USD1.16 billion individually over the 8 years. Our study provides the first long-term quantitative estimation of crops yield losses and their economic cost from surface ozone exposure in China before and after the China Clean Air Act in 2013, and improve the understanding of the spatial sensitivity of Chinese crops to ozone impacts.

## 1 Introduction

Tropospheric ozone, as a secondary air pollutant, is harmful to both human and vegetation health (Booker et al., 2009; Van Dingenen et al., 2009; Brauer et al., 2013). Since the 19$^{th}$ century, rapid industrialization and urbanization have significantly elevated the background ozone concentration in the Northern Hemisphere (The Royal Society, 2008). As a greenhouse gas that is not directly emitted by human activities, tropospheric ozone is mainly generated from sunlight driven photochemical oxidation of volatile organic compounds (VOC), carbon monoxide and methane in the presence of nitrogen oxides (NO$_X$) (Atkinson, 2000). In the past few decades, the strong linkage between fossil fuel usage and economic growth boosted emissions of ozone precursors in China. Since 2012, due to the severe fine particulate matter (PM$_{2.5}$) pollution in China, the Chinese government has adopted a stringent emission and pollution monitoring and control policy (the so-called Air Pollution Prevention and Control Action Plan (APPCAP), Zhang et al., 2016). The APPCAP has led to a significant decline of air pollutants emissions, including 17% decreases of anthropogenic emission of NO$_X$, 27% of CO, and 62% of SO$_2$ from 2010 to 2017 (Zheng et al., 2018). These significant emissions reductions have led to 33% decline of annual PM$_{2.5}$ concentration in China from 2013 to 2017, and avoided 0.41 million premature deaths associated with the ambient PM$_{2.5}$ reductions (Zhang et al., 2019). At the same time, however, anthropogenic emission of VOC increased by 11% due to the lack of effective emission controls (Zheng et al., 2018), and surface observations show that the ozone concentration in China still appears to be increasing (Wang et al., 2019; Wang et al., 2020; Li et al., 2018 & 2019a; Lu et al., 2018, 2020). The increasing trend of surface ozone may be partially explained by its decreased titration due to the decreased NO$_X$ emissions especially in megacities (Liu and Wang, 2020a, b; Li et al., 2021), or the decreasing PM$_{2.5}$ which scavenges the radical precursors of ozone (Li et al., 2019a, 2020), though this chemical pathway is still under debate (Tan et al., 2020).

The growth of ozone concentrations in China has led to emerging concerns (Lu et al., 2018, 2020). As indicated in many biological and ecological studies, high ozone concentration can seriously damage vegetation and substantially impair crop yield, which leads to economic costs and threatens food security (Krupa et al., 1998; Mills et al., 2007; Van Dingenen et al., 2009, 2018; Avnery et al., 2011a, b). Previous study estimated that for the year 2000, surface ozone exposure induced global crop yield losses of 3.9%−15% for wheat, 2.2%−5.5% for maize, and 8.5%-14.0% for soybeans, with global crop production loss of 79−121 million metric tons (Mt), based on different field-based concentration-response studies (Avnery et al., 2011a). For Eastern Asia, the ozone-induced maize reduction loss was around 3.8%, 17% for wheat, and as high as 21% for soybean in 2000 (Avnery et al., 2011a). Throughout China specifically, exposure to surface ozone in 2020 was estimated to decrease production of wheat (including both winter and spring wheat) by 6.4%−14.9% (Tang et al., 2013). Zhang et al. (2017) estimated that the current O$_3$ level in 2014 could cause annual soybean yields loss of 23.4%−30.2% in Northeast China. By using surface ozone concentration simulated from a regional chemical transport model, Lin et al. (2018) estimated that exposure to surface ozone in 2014 induced relative yield losses between 8.5% and 14% for winter wheat, 9–15.0% for rice, and 2.2–5.5% for maize, and then could cause 78.4 million metric tons of production losses from all crops. By using observational data, the

exposure to surface ozone in the North China Plain (NCP) was estimated to cause annual average of 2.3 billion USD loss for maize, and 9.3 billion USD loss for wheat from 2014 to 2017 (Feng et al., 2020; Hu et al., 2020).

To date, very few studies have investigated the long-term trends and spatial patterns of ozone impacts on crop production in China. Previous studies have been mainly focusing on a specific region of China, such as NCP (Zhang et al., 2017; Hu et al., 2020; Feng et al., 2020), or Yangtze River Delta (Wang et al., 2012). In this study, we focus on the long-term ozone-exposure impact analysis from 2010 to 2017 in China to assess the yield losses of four major crops (wheat, maize, rice, and soybean) and evaluate their associated economic losses. The specific period of 2010-2017 was chosen to cover the emission changes before and after the establishment of the APPCAP in 2013. Previous studies have been reporting the crop yield losses in one year (e.g., Lin et al., 2018; Yi et al., 2018; Feng et al., 2019a,b), or several years after the APPCAP (Zhao et al., 2020; Wang et al., 2022). Our study aims to present a comprehensive analysis of ozone-induce crop yield losses and economic impacts in the agriculture sector before and after the China APPCAP. Such an analysis is expected to provide scientific support to policymakers for their decision making.

## 2 Methodology

### 2.1 Model simulated hourly ozone and surface observation in China

Hourly ozone concentration over China from 2010 to 2017 were simulated by using a state-of-the-art global chemistry model (CAM_Chem, Lamarque et al., 2012). The original model was run at a horizontal resolution of 1.9º ×2.5º (Zhang et al, 2016; 2020, 2021), and then regridded to 1º×1º to match the crop production data (see section 2.2). The anthropogenic emissions in China from 2010 to 2017 are from the Multi-resolution Emission Inventory (MEIC) developed by Tsinghua University (http://meicmodel.org/, last access July 15, 2020). Emissions outside of China are from the Community Emissions Data System (CEDS) which were prepared for the Coupled Model Intercomparison Project Phase 6 (CMIP6) experiments (Hoesly et al., 2018). Hourly surface ozone data simulated by the model were saved from 2010 to 2017. We then adjusted the model simulated surface ozone from lowest grid box height (usually above 30 meters) to the crop height (usually 1–3 meters at the ambient observation sites), which usually reduced the simulated ozone concentration by 30-50% (Van Dingenen et al., 2009; Zhang et al., 2012).

We first evaluated the model's performance by comparing the model simulated annual average maximum daily 8-hr average (MDA8) $O_3$ with the surface observation from 2013 to 2017, which were downloaded from National Environmental Monitoring Center (CNEMC) Network (http://106.37.208.233:20035/). CNEMC collects at least 100 million environmental monitoring data from 1497 established air quality monitoring stations annually for national environmental quality assessment (Lu et al., 2018, 2020). Ozone observation data before 2013 were not available. In general, our model captures spatial patterns of the ozone distribution in China (Fig. S6 in Zhang et al., 2021), but overestimates the annual MDA8 $O_3$ concentration, with

mean bias of 5.7 ppbv and normalized mean bias of 13.7% for 5-year average from 2013 to 2017 (Table 1 in Zhang et al., 2021).

## 2.2 Ozone crop metrics

In order to assess the crop yield loss from exposure to surface ozone, different crop-ozone metrics were developed to measure the chronic ozone exposure risk of vegetation (e.g., Mauzerall and Wang, 2001; Van Dingenen et al., 2009; Avnery et al., 2011a, b). In this study, we adopted the ozone metric of AOT40 which is the European standard for the protection of vegetation, and also the commonly used and reliable indicator in both America and Asia for crop yield assessment (Tang et al., 2013; Lefohn et al., 2018; Lin et al., 2018; Feng et al., 2019a,b). The AOT40 metric is also considered as more accurate at high levels

of ozone concentration (Tuovinen, 2000; Hollaway et al., 2012; Lin et al., 2018), such as China (Lu et al., 2018, 2020). AOT40 is calculated by summing up hourly ozone exposure concentrations over the threshold of 40 ppbv during the 12 hour, 08:00— 19:59 China Standard Time (equation 1). By including concentrations over 40ppbv, AOT40 is able to sensitively capture the influence of extremely high ozone concentration (Van Dingenen et al., 2009; Hollaway et al., 2012). In a synthesis study by Mills et al. (2007), the AOT40 showed a statistically significant relationship with many crops.

$$AOT40 = \sum_{i=1}^{n}([O_3]_i - 0.04), for\ [O_3]_i \geq 0.04\ ppm \tag{1}$$

In equation 1 above, $[O_3]_i$ denotes the hourly ozone concentration level during daylight hours (8:00am – 7:59pm, GMT+8) at each grid cell ($i$), $n$ is the total hours of growing season which was counted as the 3-month harvest season based on the crop calendar (Lin et al., 2018), or 75 days composed by 44 days and 31 days before and after the anthesis dates (Feng et al., 2019a,b, 2020). Growing seasons for major crop in China were indicated in Table 1, and acquired from Major World Crop

Areas and Climate Profiles (MWCACP), and the Food and Agriculture Organization of the United Nation (FAO) (Lin et al., 2018; Zhao et al., 2020). In this study, we focused on four major crops in China—wheat, rice (including double early rice, double late rice, and single rice), maize (including north maize and south maize), and soybean.

## 2.3 Crop relative yields and economic losses

In our study, relative yield (RY) was calculated based on the exposure-response function (ERF) provided by Mills et al. (2007),

where RY is in % and AOT40 is in ppm h (Table 1). The ERF for north maize and south maize is the same with different growing seasons and growing areas (Table 1; NBSC, 2018; Zhao et al., 2020). The same ERF applies to all the rice varieties, with differences in growing season and provinces as well (Table 1).

The relative yield loss ($RYL_i$) in each grid cell was then calculated using equation 2 below.

$$RYL_i = 1 - RY_i \tag{2}$$

The crop production loss ($CPL_i$) was then calculated using equation 3 (Wang and Mauzerall, 2004; Dingenen et al., 2009; Avnery et al., 2011a,b):

$$CPL_i = CP_i \times \frac{RYL_i}{1 - RYL_i} \qquad \qquad (3)$$

where $CP_i$ is the annual crop production with unit of 1000 metric tons per year. Grid cell annual crop production data for major crops was originally developed by Van Dingenen et al. (2009) from USDA national and regional production numbers and

Agro-Ecological Zones suitability index. It contains global crop production data in 2000 with horizontal resolution of 1°×1°. We then scaled the annual national total crop yields in China to match the yearly data from the Statistical Yearbook of China from 2010 to 2017 (http://www.stats.gov.cn/tjsj/ndsj/2019/indexeh.htm, last accessed December 9th, 2021). For provinces growing both double and single Rice (such as Zhejiang and Jiangxi, see the highlighted bold notes in Table 1), the fractions for double and single rice production were scaled based on the production data from the National Bureau of Statistics

(http://data.stats.gov.cn,last accessed 26th, March, 2020). For provinces growing double early/late rice only, the rice productions were assumed to be equal for each rice (Zhao et al., 2020).

National average relative yields loss ($ARYL$, unit of %), is then calculated based on $CPL_i$ and $CP_i$ to identify the fractions of production loss in total crop production (equation 4):

$$ARYL_i = \frac{CPL_i}{CP_i + CPL_i} \times 100\% \qquad \qquad (4)$$

National economics costs for each crops ($EC_p$) were then quantified by multiplying the market price in each year using equation 5:

$$EC_p = CPL_p \times Crop\ Price_p \qquad \qquad (5)$$

where $Crop\ Price_p$ stands for the annual market price for each crop in China in every year with unit of $USD$/Mt. Crops market prices were acquired from the FAOSTAT (http://www.fao.org/faostat/, last accessed 26th, March, 2020; Feng et al., 2019a),

and shown in Table S1. The market price in 2017 is only available for maize, so we used the 3-year average from 2014 to 2016 to calculate the economic losses in 2017 for the other three crops. From 2010 to 2017, soybean has the highest crop market price, ranging from 677.9 USD/ton to 869.7 USD/ton, followed by rice (296.6 USD/ton to 559.9 USD/ton), wheat (279.5 USD/ton - 391.4 USD/ton) and maize (252.2 USD/ton–489.1 USD/ton). For soybean, wheat, and maize, the market price usually peaks in 2014 or 2015, which contributed to the peak economics loss in these years (see section 3.3).

**3 Results**

**3.1 Temporal and spatial distribution of accumulated ozone change**

Since the surface ozone in China has a distinct seasonal variations, thus making the direct comparison of the accumulated AOT40 values between the four crops impossible (Table 1), here we present the temporal and spatial distribution of annual accumulated AOT40 in China from 2010 to 2017. From Fig. 1 we see that the 8-year average annual accumulated AOT40

values are usually larger than 16 ppm h, with hotspots identified in West China (40-56 ppm h), Beijing-Tianjin-Hebei (a.k.a.

JJJ, 32-40 ppm h), Northeast China (24-32 ppm h) as well as Yangtze Delta Region (a.k.a. YRD, Fig. S1). At province level, Xizang (41.47 ppm h for 8-year average), Tianjin (34.79 ppm h) and Qinghai (34.51 ppm h) are the top three provinces with the highest annual accumulated AOT40 (Table S2). For the temporal changes, we conclude that the national annual accumulated AOT40 increased from 21.98 ppm h in 2010 to 23.85 ppm h in 2017, peaking in 2014 with 35.69 ppm h (Fig. 2).

The peak value of the annual accumulated AOT40 in 2014 as well as in 2015 were mainly caused by the high ozone values in the western China (Fig. S1), dominantly resulting from the transboundary transport from foreign sources driven by strong westerly winds and stratosphere-troposphere exchange (Zhang et al., 2008; Wang et al., 2011; Ni et al., 2018; Lu et al., 2019).

### 3.2 Growing-season ozone concentration and relative yield loss (RYL)

The accumulated AOT40 values vary among the four crops, mainly determined by the seasonality of ozone concentrations.

During the growing season for the wheat (March, April and May), the AOT40 values revealed to be higher in the Tibet Plateau and YRD, and lower in the South China, such as Hainan, Guangdong and Guangxi (Fig. 3a; Table S3). At province level (Table S3), the highest AOT40 is in Xizang (14.99 ppm h), following by Yunan (12.60 ppm h) and Qinghai (11.77 ppm h), with the lowest values in Hainan (4.43 ppm h). We also notice that the AOT40 values for wheat were decreasing in the West China, but increasing in the middle and East China from 2010 to 2017 (Fig. S2), which were caused by a combination of

unfavourable meteorological conditions and decreased anthropogenic emissions (Liu and Wang, 2020a). The RYL for wheat has the similar spatial pattern as AOT40, highest in Xizang (25.14%), and lowest in Hainan (8.13%) for the 8-year average (Table S3).

The AOT40 values for double early rice (May, June, July) are lower than those for wheat (Fig. 3b), with highest province in Anhui (12.09 ppm h), and lowest in Hainan (1.84 ppm h; Table S4). The AOT40 values for double late rice are much lower

than the early rice (Fig.3c), with highest in Fujian (6.47 ppm h), and lowest in Yunan (2.86 ppm h; Table S4). The spatial distribution of AOT40 for the early and late rice varies as well, with the hotspots in East China for early rice and South China for late rice (Figs. S3-S4). The RYLs for the double rice range from 10.71% in Anhui to 7.11% in Yunan for the 8-year average (Table S4). For the single rice, the NCP experiences high ozone exposure during the growing season (AUG, SEP, OCT), and lower in South China (Fig. 3d). AOT40 level ranges from 1.0 ppm h (Yunnan in 2017) to 14.1 ppm h (Tianjin in 2015). Highest

RYLs for the single rice are identified in the NCP including Tianjin (8-year average RYL of 10.22%), Shanxi (9.81 %), and Henan (9.67%; Table S5).

Hotspots for the AOT40 values for the north maize during growing season (JUNE, JULY, AUG) are also identified in the NCP (Fig. 3e; Fig. S7), including Tianjin (20.24ppm h), Beijing (17.92 ppm h), and Hebei (17.80 ppm h). The provincial averaged RYLs range from 0.48% in Qinghai to 5.29% in Tianjin (Table S6). When looking at the AOT40 values for the south maize

(AUG, SEP, OCT), we found that they are much lower than the north maize, with the highest in Jiangsu (8.18 ppm h for 8-year average) and lowest in Yunan (2.02 ppm h). For the spatial pattern, the AOT40 values are higher in the West China, and lower in the South China (Fig. 3d; Fig. S7). The 8-year averaged RYLs for each province are all below 1% (Table S6).

The growing seasons for soybean are the same as north maize, and thus they have the same hotpots as in NCP (Fig. 3e). For the 8-year average, the highest AOT40 values is in Tianjin (20.2 ppm h with RYL of 21.48%), followed by Beijing (17.9 ppm h with RYL of 18.79%), and Hebei (17.8 ppm h with RYL 18.65%) (Table S7).

### 3.3 Crop production loss (CPL)

From equation 3, we expect that the spatial distribution of CPL among the four crops would be different from their RYLs. From the Statistical Yearbook of China, the national wheat production increased from 115.19 Mt in 2010 to 134.34 Mt in 2017, which are mainly planted in the NCP (http://www.stats.gov.cn/tjsj/ndsj/2019/indexeh.htm, last accessed December 9[th], 2021). We estimated that on average, 26.42 Mt of wheat were lost in China due to surface ozone exposure, ranging from 13.81 Mt in 2010 to 36.51 Mt in 2015 (Fig. 4; Table S8). Fig. 5 shows the top 5 provinces with the highest wheat CPLs from 2010 to 2017, including Henan (5.23 Mt for 8-year average), Shandong (4.77), Hebei (2.79), Jiangsu (2.66), and Anhui (2.58). We conclude that the wheat CPLs due to ozone exposure were increasing from 2010 to 2017, with the peak year varying from province to province, but generally later than 2014 (Table S8). The hotspots for the wheat product losses were in the NCP, inconsistent with the patterns of RYL (Fig. S2), which was not surprising since the regions with high RYL, such as Xizang and Xingjiang in west China, usually have limited wheat productions.

National double rice (including both early and late rice) production ranges from 77.94 Mt to 80.70 Mt from 2010 to 2017. We estimated that the national CPLs for the early rice were between 3.51 Mt and 3.92 Mt in China with 8-year average of 3.60 Mt (Fig. 3; Table S9). The CPLs for the late rice were comparable with the early rice, ranging from 3.21 to 3.93 with 8-year average of 3.61 Mt (Fig. 3; Table S10). The CPLs for double early and late rice both peaked in 2014, but with different years for the lowest values (Tables S9 and S10), highlighting the seasonal variations of $O_3$ concentration between different growing season (Table 1). In China, there are more provinces (27) growing single rice than the double rice. The CPLs for the single rice were also higher than the double rice, ranging from 10.00 to 12.42 Mt, with 8-year average of 11.37 Mt (Fig. 3; Table S11). The leading provinces with the highest CPL for the single rice were Anhui (1.69–2.14 Mt from 2010 to 2017), Jiangsu (1.55–1.91), Hubei (1.13–1.52), and Sichuan (1.13–1.52), which are all exceeding 1 Mt for the 8-year average (Fig. 5; Table S11). CPLs for most provinces were shown to peak in 2015 (Fig. 6). The annual CPLs for all the rice range from 16.89 Mt in 2010 to 20.03 million Mt in 2014, with 8-year average of 18.58 Mt.

Maize in China were mainly planted in NCP and Northeast China, with north maize production dominating the total maize production (82% of the national maize productions, with the peak value of 8.17 Mt in year 2015, and lowest value of 4.59 Mt in 2017). The CPLs for maize peaked in 2015 (8.17 Mt, Fig. 4), with largest contributions from Hebei (1.02–1.81 Mt), Shandong (0.81–1.31 Mt), and Henan (0.53–0.85 Mt) (Table S12; Fig. S9). Soybean mainly grows in the NCP as well. The total soybean production in China decreased from 15.08 Mt in 2010 to 11.79 Mt 2015, and then increased slightly to 13.13 Mt in 2017. Heilongjiang was the largest soybean producer which contributed around 50% of the national soybean production (8-year average of 6.38 Mt), followed by Anhui (1.25 Mt) and Henan (1.03 Mt). We estimated that the ozone-induced CPL for

soybean ranged from 1.09 Mt in 2017 to 1.84 Mt in 2010, with 8-year annual average of 1.52 Mt (Fig. 4; Table S13). Heilongjiang, Anhui, and Henan were the three provinces with the highest CPLs, with 0.69, 0.17, 0.16 Mt losses on average individually (Table S13).

## 3.4 National average relative yield loss (ARYL) and economic loss (EC)

From Fig. 7, we conclude that wheat has the largest national ARYL, ranging from 11% to 22% from 2010 to 2017, comparable
with previous estimates: 14% in 2015 estimated by Lin et al. (2018) using regional model simulation, and 20.1-33.3% from 2015 to 2018 estimated by Zhao et al. (2020) using nationwide ozone monitoring data. National ARYLs for rice were around 8%−9%, ranking in the second place among the four crops. Lowest ARYLs were observed for south maize (0-1%). It is noteworthy that, for most crops, the highest national ARYLs were observed in 2014, while the lowest values were observed in 2010 for wheat and 2017 for other three crops (Fig. 7).
When converted to EC, we estimated that 3.86 billion to 14.29 billion USD would be lost annually due to surface ozone exposure on wheat (Fig. 8a), with the top five provinces all above 1 billion USD, including Henan, Shandong, Hebei, Jiangsu and Anhui (Table S8). Our estimates are consistent with previous study, which reported 10.3 billion and 10.7 billion USD for wheat in 2015 and 2016 (Feng et al., 2019a). National economic loss for double early and late rice increased consistently from 2.05 billion USD to 3.87 billion USD (Fig. 8a). The top three provinces with highest losses were Hunan, Jiangxi, and
Guangzhou, with 8-year average of 0.80 billion, 0.79 billion and 0.47 billion losses individually. For the single rice, national economic loss ranged from 2.96 billion to 6.49 billion USD (Fig. 8a), with top provinces in Anhui (8-year average 0.87 billion USD), Jiangsu (0.80 billion USD) and Hubei (0.63 billion USD). The ECs for the north maize ranged from 1.15 billion to 3.33 billion, much higher than the south maize (Fig. 8b). Soybean had the lowest economic losses compared with the other three crops, ranging from 0.82 billion to 1.43 billion USD annually (Fig. 8b), with the major contributions from Heilongjiang
province (335−646 million USD).

## 4 Discussions

Surface ozone emerged as an important environmental issue in China, and both modelling and observation data showed that ozone has been increasing in major megacities for the past few years (Lu et al., 2018, 2019; 2020; Li et al., 2020; Liu and Wang, 2020a,b; Ni et al., 2018; Wang et al., 2020), though strict clean air regulations have been implemented after 2013.
Exposure to high concentrations of surface ozone not only poses threat to human health, but also causes damages to crop. Our study presented a comprehensive analysis on the impact of surface ozone exposure on four major crop production loss in China, including wheat, rice (double early and late rice, single rice), maize (north maize and south maize), and soybean. Unlike the surface ozone trend, we showed that the national crop yields for major crops in China usually peaked in 2014 or 2015, shortly after the strict clean air regulations after 2013. The decreasing trend of crop yield losses from surface ozone exposure
could be explained by the fact that the surface ozone in China were increasing in urban areas, while decreasing in the rural

areas (Li et al., 2022), where the major crops are planted. Nonetheless, the relatively higher ozone, especially compared with developed countries, such as United States and Japan (Lu et al., 2018), are still posing great threats to crop productions in China. Combing the annual crop production from the Statistical Yearbook of China, we estimated that the surface ozone in China could cause an average of 26.42 million metric tons losses (Mt) of wheat production from 2010 to 2017. These losses
are even comparable to the annual average wheat production during the same period in Paris, which is the fifth largest wheat production in the world (http://www.fao.org/faostat/en/#data/QC, accessed December 12, 2021). We also estimated that the surface ozone exposure could cause 18.58 Mt losses of rice production in China, comparable to the annual rice production in Philippines, the world's 8th largest rice production. Transferring to economic values, we estimated the surface ozone exposure could cost more than 20 billion USD losses, representing more than 0.20% of annual average Gross Domestic Product (GDP)
in China from 2010 to 2017. The latest edition of the State of Food Security and Nutrition in the World estimated that between 720 and 811 million people in the world faced hunger in 2020, with 161 million increasing compared with 2019, and nearly 2.37 billion people did not have access to adequate food, with no regions spared (FAO, 2021). Therefore, reducing surface ozone pollution could not only bring the benefits of reducing ozone-related premature deaths, but also bring the benefits of control the global hunger and malnutrition issues, thus helping to reach the Sustainable Development Goal 2 of "Zero Hunger".
Meanwhile, Chinese population are projected to continue to increase, and peak around 2025 under all the shared socioeconomic pathways (SSPs, Chen et al., 2020), making it more urgent to improve the crop productions by all means.

Uncertainties exist in the design of our study, including the coarse resolution of the global transport model we used, the regional emission inventories, as we as the concentration-response functions. From the model evaluation, we learnt that our model tends to overestimate the annual MDA8 $O_3$ concentration in China. However, through sensitivity experiences, Wang et al. (2022)
showed that model biases in ozone were likely to have a relatively small impact on estimated production losses. The uncertainties from the changes in growing seasons and the concentration-response functions tend to have larger effects. We propose that further studies, using high-resolution bias-corrected ozone concentration data and region-specific response functions, need to be carried out to quantify the negative effects of surface ozone on crops. In our study, we also did not consider the possible climate changes on the crop productions. However, previous studies have demonstrated that temperature
increases could significantly reduce the crop productions as well (Asseng et al., 2015; Wiebe et al., 2015; Liu et al., 2016; Zhao et al., 2016, 2017). Despite these limitations and uncertainties, our study strives to estimate the long-term negative effects from surface ozone exposure in China before and after the clean air action in China. These estimations could provide the government and policy-makers useful references to be taken into account of the detrimental effects of ozone exposure on crop productions in China when making regional-specific ozone control policies.

**5 Conclusions and Summary**

In this study, we applied chemical transport model simulation with latest annual anthropogenic emission inventory to study the long-term trend of $O_3$-induced crop production losses from 2010 to 2017 in China. We found that the annual AOT40

(hourly ozone concentration over a threshold of 40 ppbv during the growing season) in China showed increasing trend since 2010, with peak in 2014, which was mainly caused by the high ozone concentration in West China, and then decreased

thereafter. Spatially, the annual AOT40 values were higher in West China, North China Plain, and Yangtze River Delta, with the 8-year annual average AOT40 highest in Xizang (41.47 ppm h), Tianjin (34.79 ppm h) and Qinghai (34.51 ppm h). The growing season AOT40 values were relatively higher for wheat, north maize and soybean, showing the double-hump shape for the seasonal $O_3$ distribution (with growing season of March, April and May for wheat, and June, July and August for north maize and soybean). We estimated that, at the province level, the relative yield losses (RYLs) for wheat ranged from 8.13%

to 25.14%, 6.72% to 10.71% for double early rice, 7.11% to 8.53% for double late rice, 6.79% to 10.22% for single rice, 0.48% to 5.29% for north maize, up to 0.94% for south maize, and up to 21.48% for soybean for the 8-year average from 2010 to 2017. The annual national average RYLs (ARYLs), which considers the fractions of the crop production loss with the hypothetical total production without ozone pollution, ranged from 11% to 22% for wheat, 8% to 9% for rice, 2% to 4% for north maize, ~1% for south maize, and 8.27% to 12.59% for soybean. The estimates were comparable to previous studies

(Avnery et al., 2011a; Lin et al., 2018; Feng et al., 2019a; Zhao et al., 2020; Wang et al., 2022). Using the annual crop production from the Statistical Yearbook of China, we estimated that national aggregated CPL varied from 13.81 to 36.51 million metric tons (Mt) from 2010 to 2017 for wheat. The annual CPLs for rice, including both double early and late rice, and single rice, ranged from 16.89 Mt in 2010 to 20.03 Mt in 2014. The CPLs for maize ranged from 4.59 to 8.17 Mt, and 1.09– 1.84 Mt for Soybean. Accordingly, economic losses from surface ozone exposure ranged from 3.86–14.29 billion USD for

wheat, 2.05–3.87 billion USD for double rice (with early and late rice contributes almost equally), 2.96–6.49 billion USD for single rice, 1.16–3.53 billion USD for maize (with north maize contributing to more than 90% of the total), and 0.82–1.43 billion USD for soybean.

Overall, from 2010 to 2017, the ozone-induced crop production loss in China is significant. The overlaps of major crop growing area, populated and industrial zones, and ozone concentration hotspots alerted the emergency of a better structured and balance

control on ozone precursors to limit ozone concentration and preserve food security efficiently. Our findings confirmed that the strict air quality regulations in China effectively reduced the crop yield losses associated with the high surface ozone exposure, especially in the rural areas. However, current ozone pollution in China still brings surprisingly high burdens on crop yield productions. To protect the food security in China, the government needs more efforts to control the ozone pollutions.

**Data availability.** Global anthropogenic emissions data from CEDS are available from https://www.geosci-model-dev.net/11/369/2018/ (accessed May 12th, 2021). MEIC emission inventory is available from http://meicmodel.org/?page_id=560 (last access May 6th, 2021). The CAM-Chem model is available at http://www.cesm.ucar.edu/models/cesm1.2/ (accessed May 12th, 2021). Data from CAM-Chem modelling and the processing scripts that support the findings of this study will be shared on zenodo after the paper is accepted.

**Author contributions:** YZ and DS initiated the research, and designed the paper framework. YZ ran the model, and DL processed the data, performed the data analysis, and made the plots. YZ and DL analysed the results and wrote the paper, with contributions from DD, XL and LZ.

**Competing interests:** The authors declare that they have no conflict of interest.

**Acknowledgements**: Y.Z. and D.S. acknowledge the support by the NASA GISS grant 80NSSC19M0138. We appreciate the efforts of the China Ministry of Ecology and Environment with respect to supporting the nationwide observation network and the publishing of hourly air pollutant concentrations. We also thank Prof. Qiang Zhang and Bo Zheng from Tsinghua University for providing the MEIC emission inventory in China from 2010 to 2017. We would like to thank the University of North Carolina at Chapel Hill and the Research Computing group for providing computational resources and support that have contributed to these research results. We thank Russell Harwood at Duke University for the language editing of the manuscript. We also want to express our sincere gratitude for the three reviewers who significantly improve our original manuscript.

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

**Table 1: Overview of the concentration-response function for the relative yields (RY) for ozone exposure on different crops.**

| Crops | concentration response function | Growing season |
|---|---|---|
| Wheat | RY = −0.0016×AOT40 + 0.99 | MAR, APR, MAY |
| Rice | RY = −0.0039×AOT40 + 0.94 | MAY, JUN, JUL (Double Early Rice[1]) |
| | | SEP, OCT, NOV (Double Late Rice[2]) |
| | | AUG, SEP, OCT (Single Rice) |
| Maize | RY = −0.0036×AOT40 + 1.02 | JUN, JUL, AUG (North Maize[3]) |
| | | AUG, SEP, OCT (South Maize[4]) |
| Soybean | RY = −0.0116×AOT40 + 1.02 | JUN, JUL, AUG |

[1]Double early/late rice is considered to grow in **Zhejiang, Jiangxi**, **Anhui, Hunan,** Hubei, **Fujian,** Guangdong, **Guangxi,** Hainan, **Yunnan**, Hongkong, Macao and Taiwan (Lin et al., 2019; Zhao et al., 2019).

[2]Single rice is considered to grow in Heilongjiang, Jilin, Liaoning, Hebei, Beijing, Tianjin, Shanxi, Shaanxi, Ningxia, Gansu, Xinjiang, Nei Mongol, Shanghai, Jiangsu, **Zhejiang, Anhui, Fujian, Jiangxi,** Shandong, Henan, Hubei, **Hunan, Guangxi,** Sichuan, Chongqing, Guizhou and **Yunnan** (Lin et al., 2019; Zhao et al., 2019).

[3]North maize is considered to grow in northern provinces, including Heilongjiang, Jilin, Liaoning, Beijing, Tianjin, Hebei, Henan, Shandong, Shanxi, Shaanxi, Gansu, Qinghai, Ningxia, Nei Mongol, Xinjiang and Anhui (NBSC, 2018; Zhou et al.,

545 2019).

[4]South maize are in southern provinces only, including Shanghai, Jiangsu, Zhejiang, Fujian, Jiangxi, Hubei, Hunan, Guangdong, Guangxi, Sichuan, Chongqing, Guizhou, Xizang, Yunnan, Hainan, Hongkong, Macao and Taiwan (NBSC, 2018; Zhou et al., 2019).

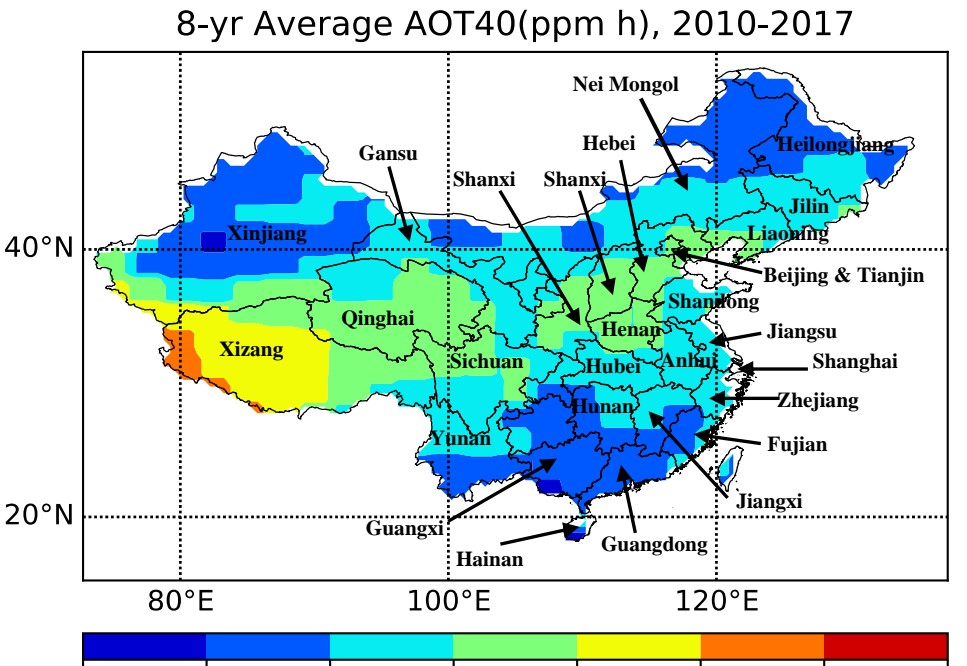

**Figure 1: Spatial distribution of annual accumulated AOT40 (ppm h) in China for 8-year average (from 2010 to 2017).**

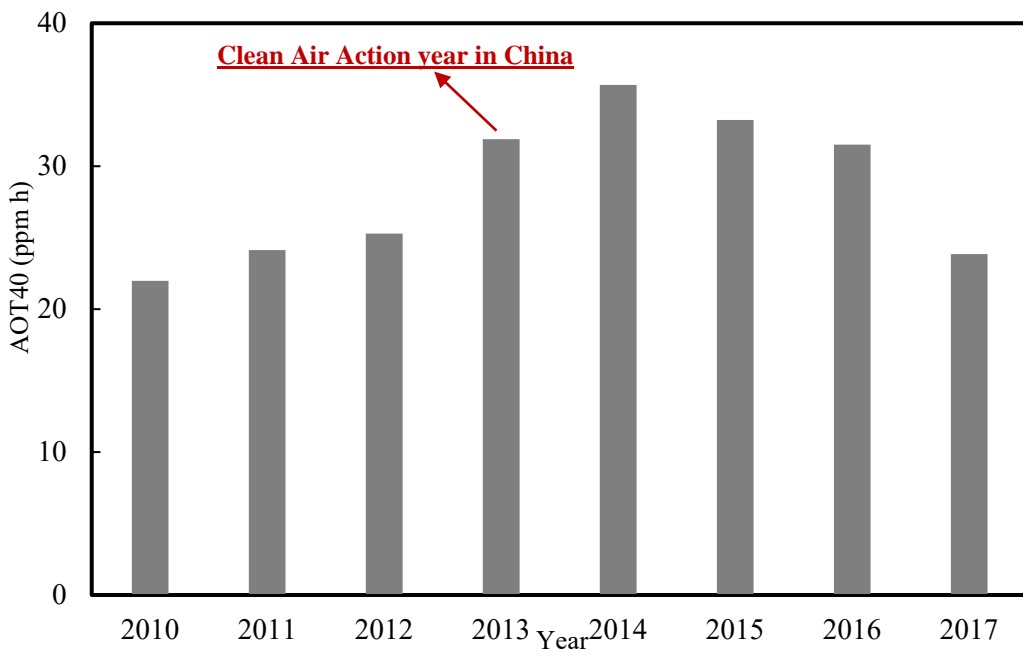

**Figure 2: The national annual accumulated AOT40 values in China from 2010 to 2017. The unit is ppm h.**

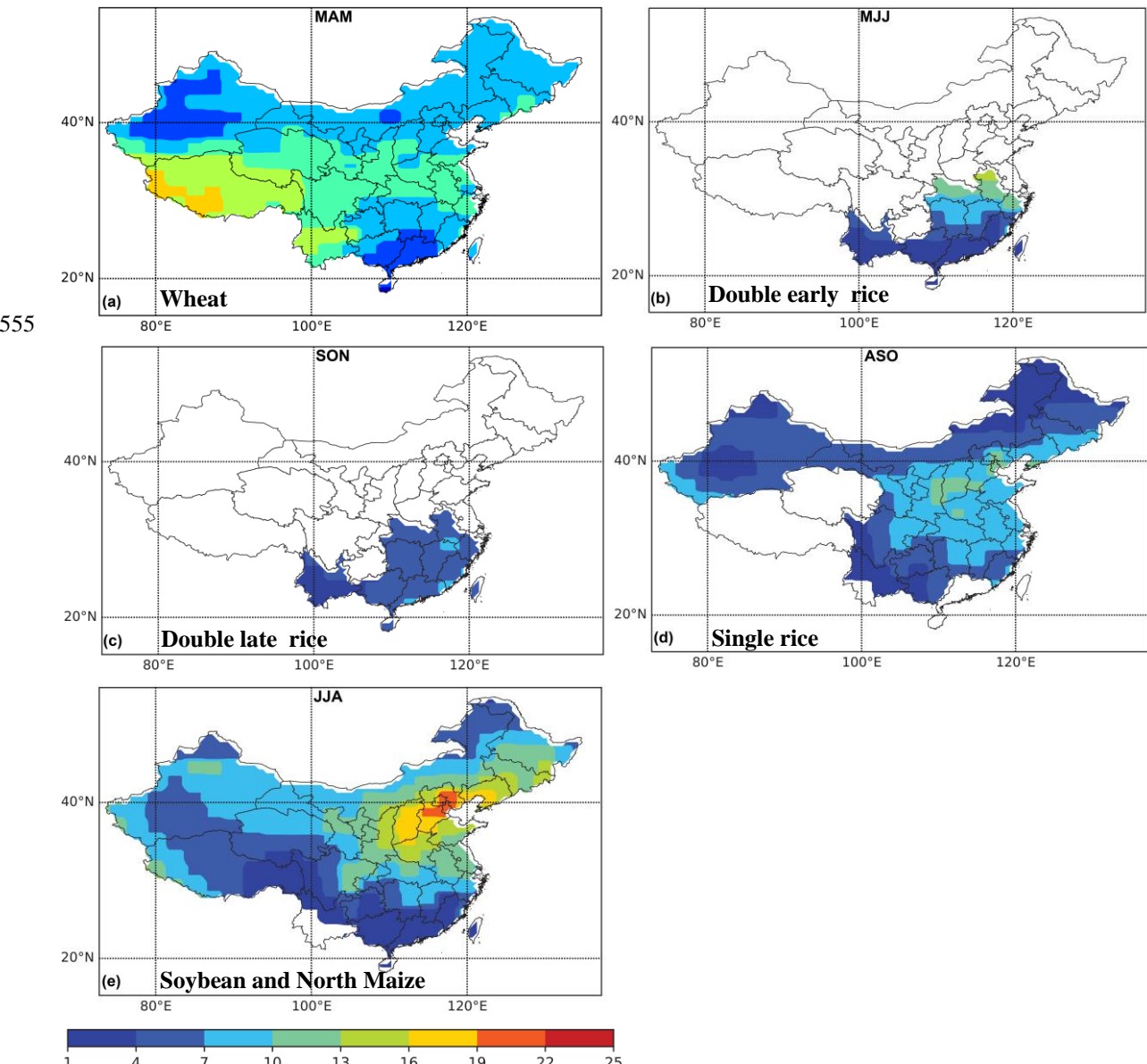


Figure 3: The spatial distribution of 8-year average growing season accumulated AOT40 for (a) wheat, (b) double early rice, (c) double late rice, (d) single rice and south maize, and (e) soybean and north maize. The unit is ppm h.


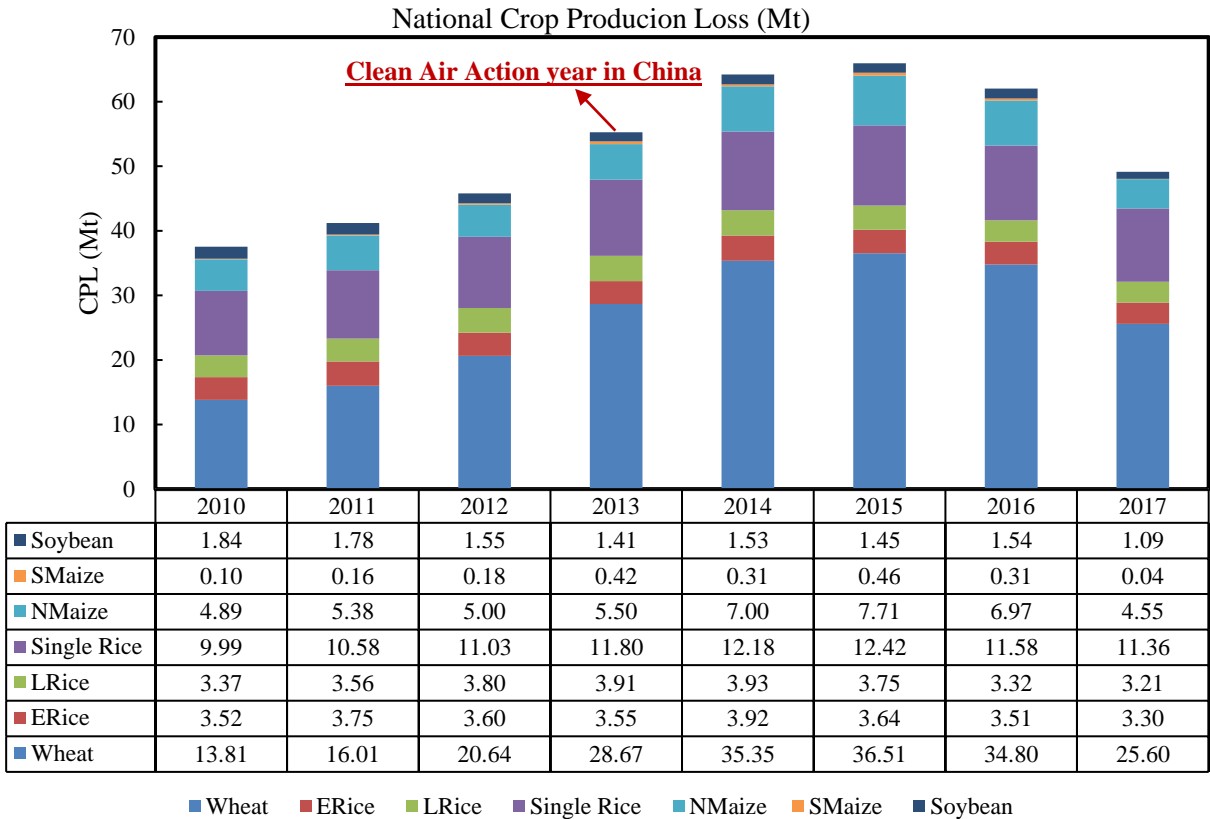

| | 2010 | 2011 | 2012 | 2013 | 2014 | 2015 | 2016 | 2017 |
|---|---|---|---|---|---|---|---|---|
| ■ Soybean | 1.84 | 1.78 | 1.55 | 1.41 | 1.53 | 1.45 | 1.54 | 1.09 |
| ■ SMaize | 0.10 | 0.16 | 0.18 | 0.42 | 0.31 | 0.46 | 0.31 | 0.04 |
| ■ NMaize | 4.89 | 5.38 | 5.00 | 5.50 | 7.00 | 7.71 | 6.97 | 4.55 |
| ■ Single Rice | 9.99 | 10.58 | 11.03 | 11.80 | 12.18 | 12.42 | 11.58 | 11.36 |
| ■ LRice | 3.37 | 3.56 | 3.80 | 3.91 | 3.93 | 3.75 | 3.32 | 3.21 |
| ■ ERice | 3.52 | 3.75 | 3.60 | 3.55 | 3.92 | 3.64 | 3.51 | 3.30 |
| ■ Wheat | 13.81 | 16.01 | 20.64 | 28.67 | 35.35 | 36.51 | 34.80 | 25.60 |

■ Wheat ■ ERice ■ LRice ■ Single Rice ■ NMaize ■ SMaize ■ Soybean

**Figure 4: National crop production loss for major crops, with SMaize for south maize, NMaize for north maize, LRice for Late Rice, ERice for early rice. Unit is million metric tons.**


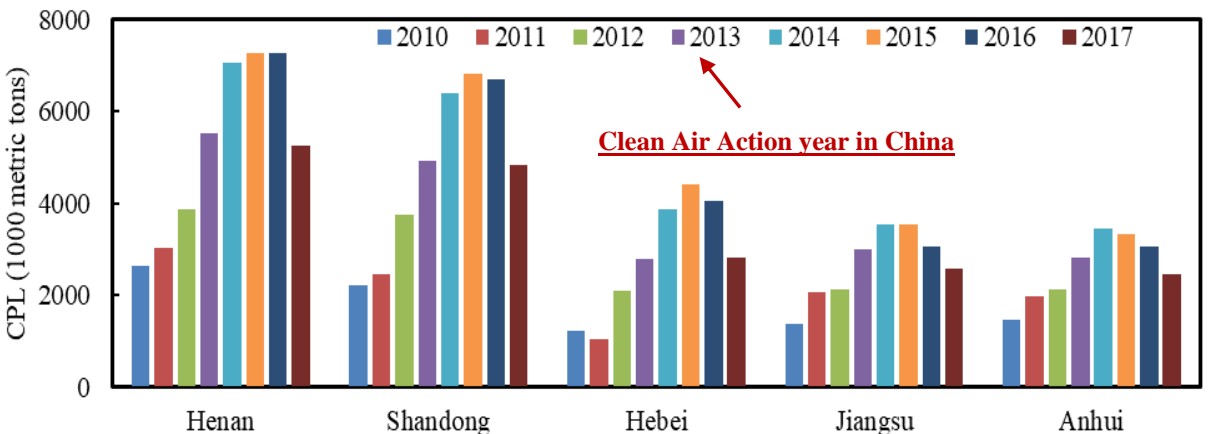

**Figure 5: Annual wheat production loss by province from 2010 to 2017 (1000 metric tons) due to surface ozone exposure.**

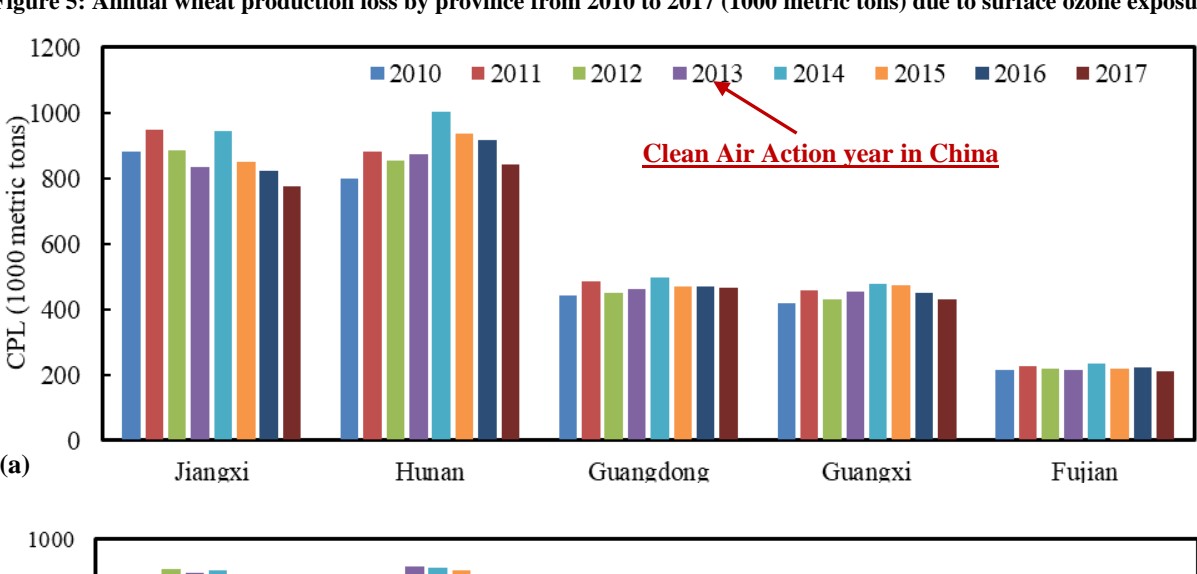

**(a)**

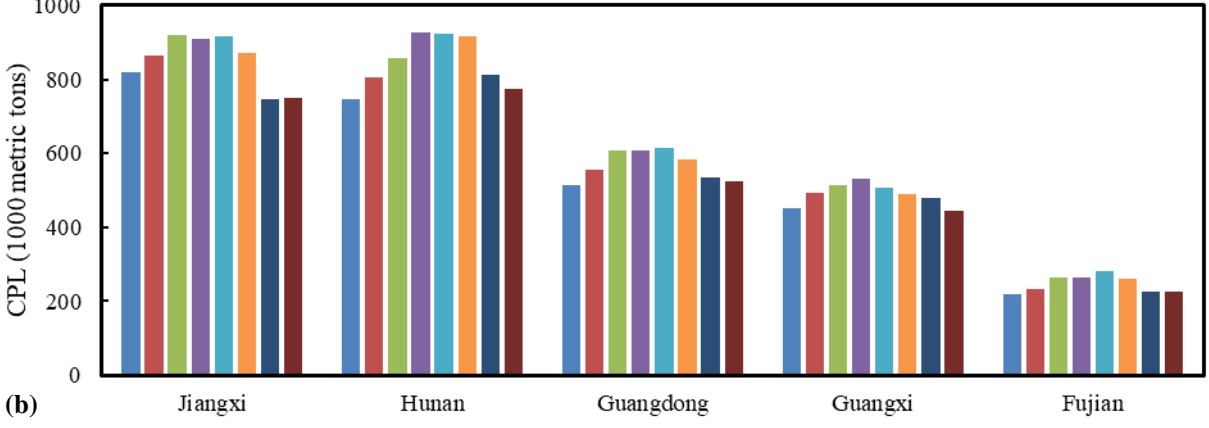

**(b)**

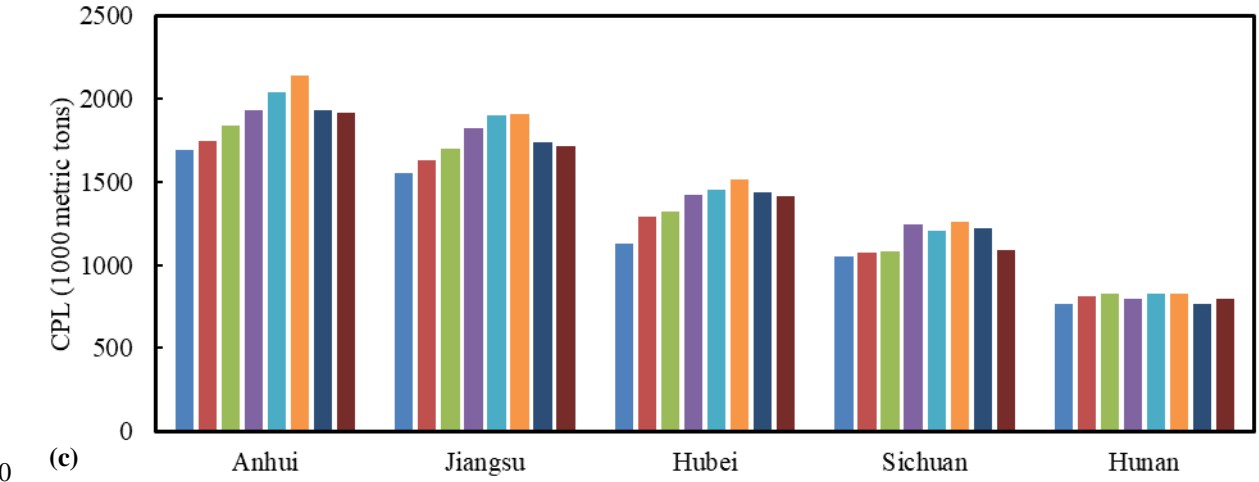

**(c)**

**Figure 6: The production losses for rice, including double early rice (a), double late rice (b), and single rice (c) in all the China provinces. Unists of thousands metric tons.**

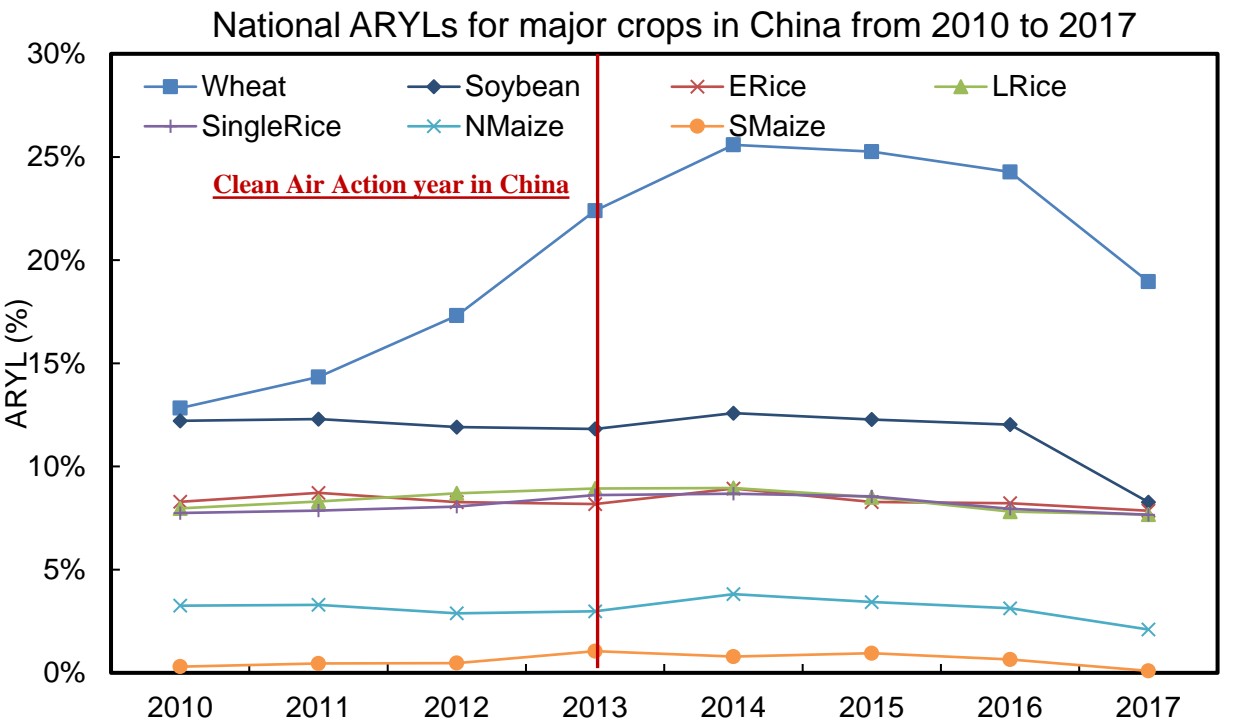

**Figure 7: National average relative yield loss (ARYLs) from 2010 to 2018. The south maize (SMaize) and soybean are multiplied by 10 to be comparable with other crops.**

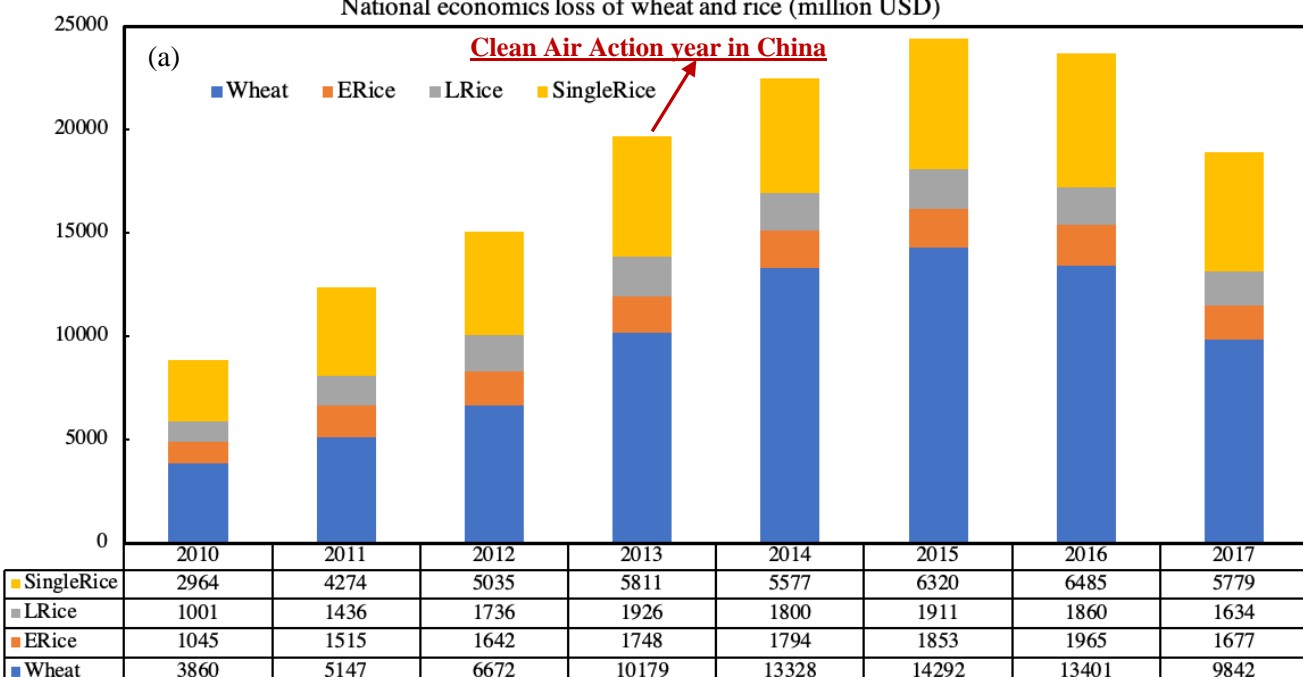

National economics loss of wheat and rice (million USD)

(a)

| | 2010 | 2011 | 2012 | 2013 | 2014 | 2015 | 2016 | 2017 |
|---|---|---|---|---|---|---|---|---|
| SingleRice | 2964 | 4274 | 5035 | 5811 | 5577 | 6320 | 6485 | 5779 |
| LRice | 1001 | 1436 | 1736 | 1926 | 1800 | 1911 | 1860 | 1634 |
| ERice | 1045 | 1515 | 1642 | 1748 | 1794 | 1853 | 1965 | 1677 |
| Wheat | 3860 | 5147 | 6672 | 10179 | 13328 | 14292 | 13401 | 9842 |

(b)

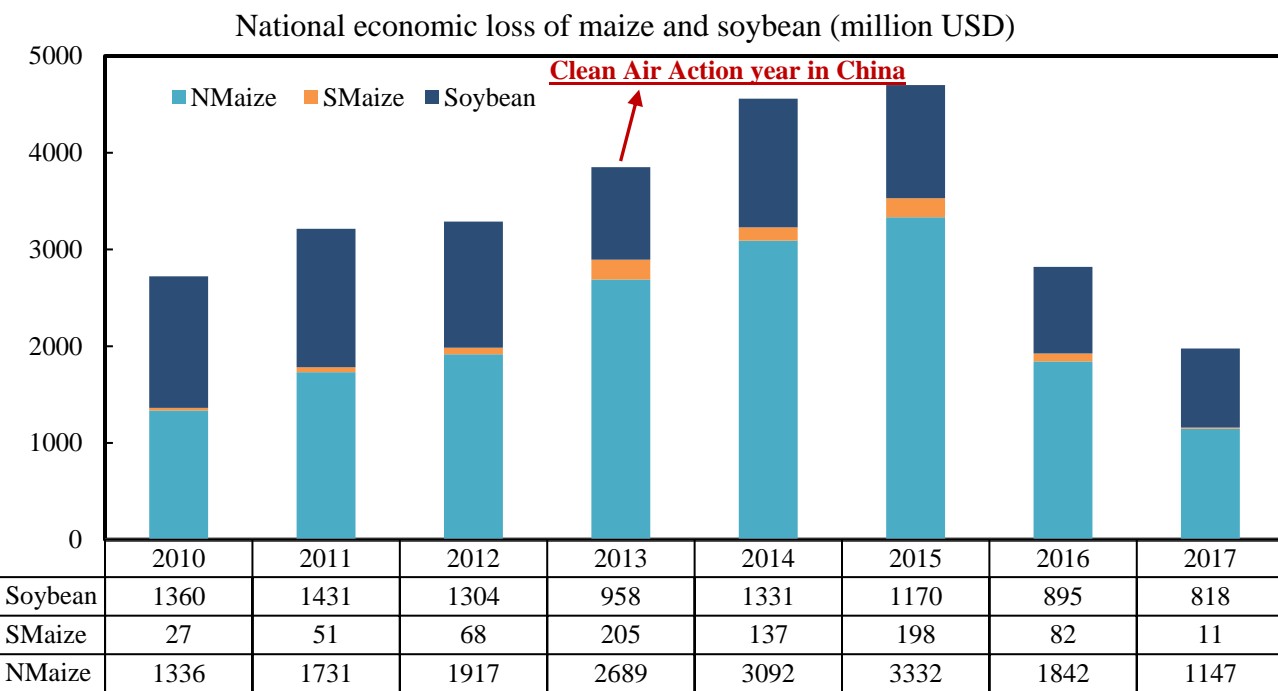

National economic loss of maize and soybean (million USD)

| | 2010 | 2011 | 2012 | 2013 | 2014 | 2015 | 2016 | 2017 |
|---|---|---|---|---|---|---|---|---|
| Soybean | 1360 | 1431 | 1304 | 958 | 1331 | 1170 | 895 | 818 |
| SMaize | 27 | 51 | 68 | 205 | 137 | 198 | 82 | 11 |
| NMaize | 1336 | 1731 | 1917 | 2689 | 3092 | 3332 | 1842 | 1147 |

**Figure 8: National economic loss from cop production loss from wheat and rice (double early & late rice, single rice) (a), and maize (north maize and south maize), and soybean (b) from 2010 to 2017. The units are million USD.**