# Peer review of "Surface ozone impacts on major crop production in China from 2010 to 2017"

_Atmospheric Chemistry and Physics, 2021_

## Author Response (AR1)

**Response to comments #1**

RC1 comments:
This study evaluated the temporal and spatial distributions of yield and economic losses due to long-term ozone exposures for major crops in China between 2010 and 2017. The results of this study are interesting, however, the novelty is not clearly stated, and the writing is very poor which requires almost a total revision. Please carefully address the comments below.
Response: We really appreciate the reviewer's efforts in providing constructive comments for our paper. We have made several modifications and implemented the suggestions as needed. We describe a few major changes, followed by our response to individual comments. We have made great efforts to work on the Results and Discussion sections. We also re-emphasized the novelty of our study in the Introduction related to the reviewer's specific question (below).

Major comments:
About the Introduction:
There are only two paragraphs in the introduction, with the first paragraph does not mention the impact of ozone on crops at all. Moreover, the scientific questions did not raise clearly in the second paragraph. Overall, I feel the introduction is not well structed, and the science question targeted in this study needs to be clearly stated.
Response: We thank the reviewer's suggestion. We now rewrite the sentences from L64-74:
"To date, very few studies have investigated the long-term trends and spatial patterns of ozone impacts on crop production in China. Previous studies have been mainly focus on a specific region of China, such as NCP (Zhang et al., 2017; Hu et al., 2020; Feng et al., 2020), or Yangtze River Delta (Wang et al., 2012). In this study, we focus on the long-term ozone-exposure impact analysis from 2010 to 2017 in China to assess the yield losses of four major crops (wheat, maize, rice, and soybean) and evaluate their associated economic losses. The specific period of 2010-2017 was chosen to cover the emission changes before and after the APPCAP established in 2013. Previous studies have been reporting the crop yield losses in one year (e.g., Lin et al., 2018; Yi et al., 2018; Feng et al., 2019a,b), or several years after the APPCAP (Zhao et al., 2019; Wang et al., 2022), and our study aims to present a comprehensive analysis of ozone-induce crop yield losses and economic impacts in the agriculture sector before and after the China APPCAP. Such an analysis is expected to provide scientific support to policymakers for their decision making."

Reference:
Feng, Z., Kobayashi, K., Li, P., Xu, Y., Tang, H., Guo, A., Paoletti, E. and Calatayud, V.: Impacts of current ozone pollution on wheat yield in China as estimated with observed ozone, meteorology and day of flowering, Atmos. Environ., 217(March), doi:10.1016/j.atmosenv.2019.116945, 2019a.

Feng, Z., De Marco, A., Anav, A., Gualtieri, M., Sicard, P., Tian, H., Fornasier, F., Tao, F., Guo, A. and Paoletti, E.: Economic losses due to ozone impacts on human health, forest productivity and crop yield across China, Environ. Int., 131(June), doi:10.1016/j.envint.2019.104966, 2019b.

Feng, Z., Hu, T., Tai, A. P. K. and Calatayud, V.: Yield and economic losses in maize caused by ambient ozone in the North China Plain (2014–2017), Sci. Total Environ., 722, 137958, doi:10.1016/j.scitotenv.2020.137958, 2020.

Hu, T., Liu, S., Xu, Y., Feng, Z. and Calatayud, V.: Assessment of O3-induced yield and economic losses for wheat in the North China Plain from 2014 to 2017, China, Environ. Pollut., 258, 113828, doi:10.1016/j.envpol.2019.113828, 2020.

Lin, Y., Jiang, F., Zhao, J., Zhu, G., He, X., Ma, X., Li, S., Sabel, C. E. and Wang, H.: Impacts of O3 on premature mortality and crop yield loss across China, Atmos. Environ., 194(July), 41–47, doi:10.1016/j.atmosenv.2018.09.024, 2018.

Wang, X., Zhang, Q., Zheng, F., Zheng, Q., Yao, F., Chen, Z., Zhang, W., Hou, P., Feng, Z., Song, W., Feng, Z. and Lu, F.: Effects of elevated $O_3$ concentration on winter wheat and rice yields in the Yangtze River Delta, China, Environ. Pollut., 171, 118–125, doi:10.1016/j.envpol.2012.07.028, 2012.

Wang, Y., Wild, O., Ashworth, K., Chen, X., Wu, Q., Qi, Y. and Wang, Z.: Reductions in crop yields across China from elevated ozone, Environ. Pollut., 292(118218), doi:https://doi.org/10.1016/j.envpol.2021.118218, 2022.

Yi, F., McCarl, B. A., Zhou, X. and Jiang, F.: Damages of surface ozone: Evidence from agricultural sector in China, Environ. Res. Lett., 13(3), doi:10.1088/1748-9326/aaa6d9, 2018.

Zhang, W., Feng, Z., Wang, X., Liu, X. and Hu, E.: Quantification of ozone exposure- and stomatal uptake-yield response relationships for soybean in Northeast China, Sci. Total Environ., 599–600, 710–720, doi:10.1016/j.scitotenv.2017.04.231, 2017.

Zhao, H., Zheng, Y., Zhang, Y. and Li, T.: Evaluating the effects of surface $O_3$ on three main food crops across China during 2015–2018, Environ. Pollut., 258, 113794, doi:10.1016/j.envpol.2019.113794, 2020.

**About the section 2.2**
Line 85-90: The authors mentioned that there are many different crop-ozone matrixes available, and they adopted AOT40 in their study. The authors should at least lay out a few crop-ozone matrixes, and talk about the possible advantage and disadvantage among different matrixes. At the end, give a reason why AOT40 is selected.
Response: AOT40 metric is the European standard for the protection of vegetation, and widely used in both America and Asia (Tang et al., 2013; Lefohn et al., 2018; Lin et al., 2018). The AOT40 metric is also considered as more accurate at high levels of ozone concentration (Tuovinen, 2000; Hollaway et al., 2012), which is the case for ozone pollution in China (Lu et al., 2018, 2020). To clarify this, we modify the sentence in line 97:

"In this study, we adopted the ozone metric of AOT40 which is the European standard for the protection of vegetation, and also the commonly used and reliable indicator in both America and Asia for crop yield assessment (UNECE, 2017; Tang et al., 2013; Lefohn et al., 2018; Lin et al., 2018; Feng et al., 2019a,b). The AOT40 metric is also considered as more accurate at high levels of ozone concentration (Tuovinen, 2000; Hollaway et al., 2012), which is the case for China (Lu et al., 2018, 2020)."

Reference:
Hollaway, M. J., Arnold, S. R., Challinor, A. J., and Emberson, L. D.: Intercontinental trans-boundary contributions to ozone-induced crop yield losses in the Northern Hemisphere, Biogeosciences, 9, 271–292, https://doi.org/10.5194/bg-9-271-2012, 2012.

Lefohn, A. S., Malley, C. S., Smith, L., Wells, B., Hazucha, M., Simon, H., Naik, V., Mills, G., Schultz, M. G., Paoletti, E., De Marco, A., Xu, X., Zhang, L., Wang, T., Neufeld, H. S., Musselman, R. C., Tarasick, D., Brauer, M., Feng, Z., Tang, H., Kobayashi, K., Sicard, P., Solberg, S. and Gerosa, G.: Tropospheric ozone assessment report: Global ozone metrics for climate change, human health, and crop/ecosystem research, Elementa, 6, doi:10.1525/elementa.279, 2018.

Lu, X., Zhang, L., Chen, Y., Zhou, M., Zheng, B., Li, K., Liu, Y., Lin, J., Fu, T.-M., and Zhang, Q.: Exploring 2016–2017 surface ozone pollution over China: source contributions and meteorological influences, Atmos. Chem. Phys., 19, 8339–8361, https://doi.org/10.5194/acp-19-8339-2019, 2019.

Lu, X., Zhang, L., Wang, X., Gao, M., Li, K., Zhang, Y., Yue, X. and Zhang, Y.: Rapid Increases in Warm-Season Surface Ozone and Resulting Health Impact in China Since 2013, Environ. Sci. Technol. Lett., doi:10.1021/acs.estlett.0c00171, 2020.

Tang, H., Takigawa, M., Liu, G., Zhu, J., & Kobayashi, K. (2013). A projection of ozone-induced wheat production loss in China and India for the years 2000 and 2020 with exposure-based and flux-based approaches. Global Change Biology, 19(9), 2739-2752.

Tuovinen, J. P.: Assessing vegetation exposure to ozone: Properties of the AOT40 index and modifications by deposition modelling, Environ. Pollut., 109(3), 361–372, doi:10.1016/S0269-7491(00)00040-3, 2000.

UNECE, 2017. Chapter 3: mapping critical levels for vegetation. International cooperative programme on effects of air pollution on natural vegetation and crops. Bangor, UK. http://icpvegetation.ceh.ac.uk/, last accessed 24 November 2021.

Line 94-95: The authors should specifically define or point out the crops corresponding to what growing season, as multiple growing seasons were mentioned in the manuscript.
Response: We thank the reviewer pointing this out. We now add the following sentence to point the readers to the growing season for different crops:
"Growing seasons for major crops in China were indicated in Table 1, and acquired from Major World Crop Areas and Climate Profiles (MWCACP), and the Food and Agriculture Organization of the United Nation (FAO) (Lin et al., 2018; Zhao et al., 2020)."

Line 96-97: Are there any differences in the definition of the growing season between NWCACP and FAO? The authors only mentioned NWCACP and FAO with no details.
Response: The definitions of the growing season between NWCACP and FAO are the same.

**About the section 3.3**
There are redundant descriptions. For instance, the authors made some comparison to some literature, i.e., Lines 180, 181, 199, 205 and 206. However, they repeat similar discussions later (i.e., Line 270-273). In addition, the comparison does not specifically mention what year or period, making the comparison invalid.
Response: We thank the reviewer pointing this out. We now delete the comparisons between the crop yields losses in China with the annual crop productions in other countries in section 3.3. Instead, we only kept them at the discussions line 249-line 254:
"Combing the annual crop production from the Statistical Yearbook of China, we estimated that the surface ozone in China could cause an average of 26.42 million metric tons losses (Mt) of

wheat production from 2010 to 2017. These losses are even comparable to the annual average wheat production during the same period in Paris, which is the fifth largest wheat production in the world (http://www.fao.org/faostat/en/#data/QC, accessed December 12, 2021). We also estimated that the surface ozone exposure could cause 18.58 Mt losses of rice production in China, comparable to the annual rice production in Philippines, the world's 8[th] largest rice production."

Line 191-193
The authors highlight the differences in the calculations of the growing season lead to different years for the lowest values. In my opinion, this is nothing new worth highlighting.
Response: We agree with the reviewer that we should not highlight the calculations of different growing seasons. Instead, we revised the sentence in line 192, and highlighted the seasonality of $O_3$ concentration:
"The CPL for double early and late rice both peak in 2014, but with different years for the lowest values (Tables S9 and S10), highlighting the seasonal variations of $O_3$ concentration between different growing seasons (Table 1)."

Line 212-213:
It seems to be contradictory that the authors stated the lowest CPL in northeast China, and then emphasize Heilongjiang is the highest. Later on, I realize the statement of highest yield in Heilongjiang is probably in another year. The authors need to carefully check out the entire manuscript to make the statement clear and readable.
Response: We thank the reviewer pointing this out. We now rewrite this sentence to avoid confusion in line 216-217:
"We estimated that the ozone-induced CPL for soybean ranges from 1.09 Mt in 2017 to 1.84 Mt in 2010, with 8-year annual average of 1.52 Mt (Fig. 4; Table S13). Heilongjiang, Anhui, and Henan are the three provinces with the highest CPL, with 0.69, 0.17, 0.16 Mt loss on average individually (Table S13)."

**The Discussions section needs a total revision.**
Response: We appreciate the reviewer's comment about the discussion. We now rewrite this part. In the new discussion section, we talked about the decreasing trend of ozone-induced crop yields losses in China after 2013, the future climate and population changes on crops, and also the uncertainties for our study originating from the model, the emission inventories and the concentration-response function we used. We also rewrote the Results and Summary section to show the results only.
**"4 Discussions**

Surface ozone emerged as an important environmental issue in China, and were shown increasing trend in major megacities for the past few years using both modelling and observation data (Lu et al., 2018, 2019; 2020; Li et als., 2020; Liu and Wang, 2020a,b; Ni et al., 2018; Wang et al., 2020), though strict clean air regulations have been implemented after 2013. Exposure to high concentrations of surface ozone not only poses threat to human health, but also cause damages to crop. Our study presented a comprehensive analysis on the impact of surface ozone exposure on four major crop production loss in China, including wheat, rice (double early and late rice, single rice), maize (north maize and south maize), and soybean. Unlike the surface ozone trend, we showed that the national crop yields for major crops in China usually peaks in

2014 or 2015, shortly after the strict clean air regulations after 2013. The decreasing trend of crop yield losses associated with surface ozone exposure was mainly explained by the fact that the surface ozone in China were increasing in urban areas, while decreasing in the rural areas (Li et al., 2022), where the major crops are planted.  Nonetheless, the relatively higher ozone, especially compared with developed countries, such as United States and Japan (Lu et al., 2018), are still posing great threats to crop productions in China. Combing the annual crop production from the Statistical Yearbook of China, we estimated that the surface ozone in China could cause an average of 26.42 million metric tons losses (Mt) of wheat production from 2010 to 2017. These losses are even comparable to the annual average wheat production during the same period in Paris, which is the fifth largest wheat production in the world (http://www.fao.org/faostat/en/#data/QC, accessed December 12, 2021). We also estimated that the surface ozone exposure could cause 18.58 Mt losses of rice production in China, comparable to the annual rice production in Philippines, the world's 8[th] largest rice production. Transferring to economic values, we estimated the surface ozone exposure could cost more than 20 billion $ losses, representing more than 0.20% of annual average Gross Domestic Product (GDP) in China from 2010 to 2017. The latest edition of the State of Food Security and Nutrition in the World estimated that between 720 and 811 million people in the world faced hunger in 2020, with 161 million increasing compared with 2019, and nearly 2.37 billion people did not have access to adequate food, with no regions spared (FAO, 2021). Therefore, reducing surface ozone pollution could not only bring the benefits of reducing ozone-related premature deaths, but also bring the benefits of control the global hunger and malnutrition issues, thus helping to reach the Sustainable Development Goal 2 of "Zero Hunger". Meanwhile, Chinese population are projected to continue to increase and peak around 2025 under all the shared socioeconomic pathways (SSPs, Chen et al., 2020), making it more urgent to improve the crop productions by all means.

    Uncertainties exist in the design of our study, including the coarse resolution of the global transport model we used, the regional emission inventories, as we as the concentration-response functions. From the model evaluation, we learnt that our model tends to overestimate the annual MDA8 $O_3$ concentration in China. However, through sensitivity experiences, Wang et al. (2022) showed that model biases in ozone were likely to have a relatively small impact on estimated production losses. The uncertainties from the changes in growing seasons, and the concentration-response functions tend to have larger effects. We propose that further studies, using high-resolution bias-corrected ozone concentration data and region-specific response functions, need to be carried out to quantify the negative effects of surface ozone on crops. In our study, we also did not consider the possible climate changes on the crop productions. However, previous studies have demonstrated that temperature increases could significantly reduce the crop productions as well (Asseng et al., 2015; Wiebe et al., 2015; Liu et al., 2016; Zhao et al., 2016, 2017). Despite these limitations and uncertainties, our study strives to estimate the long-term negative effects from surface ozone exposure in China before and after the clean air action in China. These estimations could provide the government and policy-makers useful references to be taken into account of the detrimental effects of ozone exposure on crop productions in China when making regional-specific ozone control policies."

There are many sentences duplicated in several places. For instance, Line 81-82, "In general, the model simulated AOT40 values were lower than the observation data, with normalized mean bias ranging from -5% in 2015 to -28% in 2017". Line 244-246, and 278-279

Response: We thank the reviewer pointing this out. We now remove the repeated sentences in 244-246, 278-279.

Lines 238-246: very wordy, should be trimmed substantially and make clear what major message the authors want to convey.
Response: We thank the reviewer's comment. We now shorten this sentence, and put them into the beginning of the section "5 Conclusions and Summary", line 277:
"In this study, we applied chemical transport model simulation with updated annual anthropogenic emission inventory to study the long-term trend of $O_3$-induced crop production losses from 2010 to 2017 in China."

The second paragraph of Discussions only lay out many results without any depth.
Response: Please see our response above. We have rewritten our Discussion and Conclusion sections. All the detailed results were put into the new "Conclusions and Summary" section, and we discussed the uncertainties in the design of our study, the possible shortage of this study, and the policy implications as indicated from this study. We believe our discussion is more clear now.

The authors mentioned many times of the year 2014 (i.e., Lines 185, 192, and 219), but what special with the year has never been mentioned.
Response: We thank the reviewer pointing this out. In our study design, the specific period of 2010-2017 was chosen to cover the emission changes before and after the China's Air Pollution Prevention and Control Action Plan (APPCAP) which was established in 2013. Studies have shown that the anthropogenic emissions for major air pollutants are seen significant decline (Zheng et al., 2018; Zhang et al., 2019) after 2013. However, the summertime ozone in China urban regions have been reported to continue to increase after 2013 (Lu et al., 2018, 2020; Li K. et al., 2019, 2020; Li X. et al., 2022). In our study, we showed that crop yield losses associated with ozone exposure generally peak before 2014, and then decrease thereafter, demonstrating the fact that the surface ozone in rural China have a decreasing trend, consistent with the long-term observations (Li X. et al., 2022).
Reference:

Li, K., Jacob, D. J., Zhang, Q., Liao, H., Bates, K. H. and Shen, L.: Anthropogenic drivers of 2013–2017 trends in summer surface ozone in China, Proc. Natl. Acad. Sci., 116(2), 422–427, doi:10.1073/pnas.1812168116, 2019a.

Li, K., Jacob, D. J., Shen, L., Lu, X., De Smedt, I., and Liao, H.: Increases in surface ozone pollution in China from 2013 to 2019: anthropogenic and meteorological influences, Atmos. Chem. Phys., 20, 11423–11433, https://doi.org/10.5194/acp-20-11423-2020, 2020.

Li, X., Yuan, B., Parrish, D. D., Chen, D., Song, Y., Yang, S., Liu, Z. and Shao, M.: Long-term trend of ozone in southern China reveals future mitigation strategy for air pollution, Atmos. Environ., 269(118869), doi:https://doi.org/10.1016/j.atmosenv.2021.118869, 2022.

Lu, X., Hong, J., Zhang, L., Cooper, O. R., Schultz, M. G., Xu, X., Wang, T., Gao, M., Zhao, Y. and Zhang, Y.: Severe Surface Ozone Pollution in China: A Global Perspective, Environ. Sci. Technol. Lett., 5(8), 487–494, doi:10.1021/acs.estlett.8b00366, 2018.

Lu, X., Zhang, L., Wang, X., Gao, M., Li, K., Zhang, Y., Yue, X. and Zhang, Y.: Rapid Increases in Warm-Season Surface Ozone and Resulting Health Impact in China Since 2013, Environ. Sci. Technol. Lett., doi:10.1021/acs.estlett.0c00171, 2020.

Zhang, Q., Zheng, Y., Tong, D., Shao, M., Wang, S., Zhang, Y., Xu, X., Wang, J., He, H., Liu, W., Ding, Y., Lei, Y., Li, J., Wang, Z., Zhang, X., Wang, Y., Cheng, J., Liu, Y., Shi, Q., Yan, L., Geng, G., Hong, C., Li, M., Liu, F., Zheng, B., Cao, J., Ding, A., Gao, J., Fu, Q., Huo, J., Liu, B., Liu, Z., Yang, F., He, K. and Hao, J.: Drivers of improved $PM_{2.5}$ air quality in China from 2013 to 2017, Proc. Natl. Acad. Sci. U. S. A., 116(49), 24463–24469, doi:10.1073/pnas.1907956116, 2019.

Zheng, B., Tong, D., Li, M., Liu, F., Hong, C., Geng, G., Li, H., Li, X., Peng, L., Qi, J., Yan, L., Zhang, Y., Zhao, H., Zheng, Y., He, K. and Zhang, Q.: Trends in China's anthropogenic emissions since 2010 as the consequence of clean air actions, Atmos. Chem. Phys., 18(19), 14095–14111, doi:10.5194/acp-18-14095-2018, 2018.

The authors mentioned spatial heterogeneity across regions and provinces, but with no details.
Response: We now delete this sentence in our new Section 5 "Conclusions and Summary".

The last few sentences talked about thess ozone pollution control over different regions. However, the major role of ozone on CPL over these regions have not been well discussed at all.
Response: We appreciate the reviewer's question. We now remove this sentence since we made great efforts to reconstruct our results and discussions and felt that these last few sentences are not necessary anymore.

**Minor comments:**
Line 59-60: This message of the sentence is not clearly stated. The sentence writes that previous studies have focused on crop production loss from ozone at the global scale. Have any of the studies focused on China?
Response: There are published studies focusing on the China crop yields loss. To avoid confusion, we now rewrite the sentences about the novelty about our study:
"To date, very few studies have investigated the long-term trends and spatial patterns of ozone impacts on crop production in China. Previous studies have been mainly focus on a specific region of China, such as NCP (Zhang et al., 2017; Hu et al., 2020; Feng et al., 2020), or Yangtze River Delta (Wang et al., 2012). In this study, we focus on the long-term ozone-exposure impact analysis from 2010 to 2017 in China to assess the yield losses of four major crops (wheat, maize, rice, and soybean) and evaluate their associated economic losses. The specific period of 2010-2017 was chosen to cover the emission changes before and after the APPCAP established in 2013. Previous studies have been reporting the crop yield losses in one year (e.g., Lin et al., 2018; Yi et al., 2018; Feng et al., 2019a,b), or several years after the APPCAP (Zhao et al., 2019; Wang et al., 2022), and our study aims to present a comprehensive analysis of ozone-induce crop yield losses and economic impacts in the agriculture sector before and after the China APPCAP. Such an analysis is expected to provide scientific support to policymakers for their decision making."

Line 85: Matrixes: should be metrics?
Response: We changed to "metrics"

Line 129: The price for each crop during 2010-2017 is given based on the min/max, however, the readers do not know the specific price corresponding to each year. A table might be useful to lay out the prices.

Response: We appreciate the question. We now add a new Table in the supporting:

Table S1. The crop market prices for the major crops in China, acquired from the FAOSTAT (unit of $ per ton; http://www.fao.org/faostat/, last accessed 26th, March, 2020).

| **Year** | 2010 | 2011 | 2012 | 2013 | 2014 | 2015 | 2016 | 2017[1] |
|---|---|---|---|---|---|---|---|---|
| Wheat | 279.5 | 321.5 | 323.3 | 355.1 | 377 | 391.4 | 385.1 | 384.5 |
| Soybean | 738.5 | 803.4 | 841.5 | 677.9 | 869.7 | 808.2 | 581.5 | 753.1 |
| Maize | 273.3 | 321.8 | 383.5 | 489.1 | 441.9 | 432.4 | 264.3 | 252.2 |
| Rice | 296.6 | 403.9 | 456.4 | 492.3 | 457.9 | 508.9 | 559.9 | 508.9 |

Line 162: RYLs for specific crops should be clearly written.

Response: We now revise this sentence:

"The RYLs for the double rice, range from 10.71% in Anhui to 7.11% in Yunan for the 8-year average (Table S4)."

Line 178: The citation of the Statistical Yearbook of China should be added.

Response: Thanks for pointing out. We now add the citation for the Statistical Yearbook of China.

"From the Statistical Yearbook of China, the national wheat production increased from 115.19 million Mt in 2010 to 134.34 million Mt in 2017, which are mainly planted in the NCP (http://www.stats.gov.cn/tjsj/ndsj/2019/indexeh.htm, last accessed December 9[th], 2021)."

Line 183: "CPL" should be replaced by "wheat CPL"

Response: we made the change following the reviewer's suggestion.

"Fig. 5 shows the wheat CPL for each province in China from 2010 to 2017"

Line 223: studies changed to study.

Response: We changed to "study".

Table S13: The production loss of China in 2017 was miscalculated and "374" should be replaced by "74". Please carefully check all the calculations in the tables.

Response: We thank the reviewer finding this out. We now change to the right number.

Line 235: References should be added.

Response: We now add the following references here:

"Exposure to high concentrations of surface ozone not only poses threat to human health, but also cause damages to crops (Krupa et al., 1998; EPA, 1996; EEA 1999; Mauzerall & Wang, 2001)."

Reference

EEA, 1999. Environmental assessment Report No. 2. Environment in the European Union at the turn of the century. European Environmental Agency, Copenhagen, 446pp.

EPA, 1996. Air quality criteria for ozone and related photochemical oxidants. United States Environmental Protection Agency (EPA), pp. 1–1 to 1–33.

Krupa, S. V., Nosal, M. and Legge, A. H.: Short communication A numerical analysis of the combined open-top chamber data from the USA and Europe on ambient ozone and negative crop responses,101, 157–160, 1998.

Mauzerall, D. L. and Wang, X.: Protecting agricultural crops from the effects of tropospheric ozone exposure: reconciling Science and Standard Setting in the United States, Europe, and Asia, Annu. Rev. Energy Environ., 26(1), 237–268, doi:10.1146/annurev.energy.26.1.237, 2001.

Line 237: The authors said the previous studies only focused on small regions. However, in the introduction, the authors mentioned there are studies with a focus of the globe. This seems to be contradictory.
Response: We thank the reviewer pointing this out. Here we meant small regions in China. There are significant number of studies focusing on global. We now rewrite this sentence to be more precise:
"Previous studies have been using modelling results or observation data to study the crop production losses in China for a single year (Lin et al., 2018; Feng et al., 2019b), or several years at specific regions, such as North China Plain (Zhang et al., 2017; Hu et al., 2020; Feng et al., 2020), or Yangtze River Delta (Wang et al., 2012). Some studies also estimated crop yield losses for three or four years in China after 2013 (Zhao et al., 2020; Wang et al., 2022), when the Chinese government implemented the stringent Air Pollution Prevention and Control Action Plan (APPCAP)."

Reference:

Feng, Z., De Marco, A., Anav, A., Gualtieri, M., Sicard, P., Tian, H., Fornasier, F., Tao, F., Guo, A. and Paoletti, E.: Economic losses due to ozone impacts on human health, forest productivity and crop yield across China, Environ. Int., 131(June), doi:10.1016/j.envint.2019.104966, 2019b.

Feng, Z., Hu, T., Tai, A. P. K. and Calatayud, V.: Yield and economic losses in maize caused by ambient ozone in the North China Plain (2014–2017), Sci. Total Environ., 722, 137958, doi:10.1016/j.scitotenv.2020.137958, 2020.

Hu, T., Liu, S., Xu, Y., Feng, Z. and Calatayud, V.: Assessment of $O_3$-induced yield and economic losses for wheat in the North China Plain from 2014 to 2017, China, Environ. Pollut., 258, 113828, doi:10.1016/j.envpol.2019.113828, 2020.

Lin, Y., Jiang, F., Zhao, J., Zhu, G., He, X., Ma, X., Li, S., Sabel, C. E. and Wang, H.: Impacts of O3 on premature mortality and crop yield loss across China, Atmos. Environ., 194(July), 41–47, doi:10.1016/j.atmosenv.2018.09.024, 2018.

Wang, X., Zhang, Q., Zheng, F., Zheng, Q., Yao, F., Chen, Z., Zhang, W., Hou, P., Feng, Z., Song, W., Feng, Z. and Lu, F.: Effects of elevated O3 concentration on winter wheat and rice yields in the Yangtze River Delta, China, Environ. Pollut., 171, 118–125, doi:10.1016/j.envpol.2012.07.028, 2012.

Wang, Y., Wild, O., Ashworth, K., Chen, X., Wu, Q., Qi, Y. and Wang, Z.: Reductions in crop yields across China from elevated ozone, Environ. Pollut., 292(118218), doi:https://doi.org/10.1016/j.envpol.2021.118218, 2022.

Yi, F., McCarl, B. A., Zhou, X. and Jiang, F.: Damages of surface ozone: Evidence from agricultural sector in China, Environ. Res. Lett., 13(3), doi:10.1088/1748-9326/aaa6d9, 2018.

Zhang, W., Feng, Z., Wang, X., Liu, X. and Hu, E.: Quantification of ozone exposure- and stomatal uptake-yield response relationships for soybean in Northeast China, Sci. Total Environ., 599–600, 710–720, doi:10.1016/j.scitotenv.2017.04.231, 2017.

Zhao, H., Zheng, Y., Zhang, Y. and Li, T.: Evaluating the effects of surface $O_3$ on three main food crops across China during 2015–2018, Environ. Pollut., 258, 113794, doi:10.1016/j.envpol.2019.113794, 2020.

Line 255: It should be written clearly whether the CPL and EL for a particular crop or the total CPL and EL for all four crops.
Response: We appreciate the reviewer's question. We now remove this sentence since we made great efforts to reconstruct our results and discussions and felt that this sentence is not necessary anymore.

**Response to comments #2**

RC2 comments:
General comments:
This is a manuscript delivering important messages towards China's air quality policymaking. They found that crop yield damages due to ozone air pollution have increased in recent years and are especially large for wheat and rice. Accumulatively, the economic losses are substantial, i.e. around ~20 billion USD for major crops during the past 8 years. Findings of this study indicate that improving China's ozone air quality can benefit food security, in addition to human health which has been the dominant driver of previous clean air policies. This reviewer works broadly in the arena of atmospheric chemistry and policy-relevant science instead of being an expert on vegetation impacts of ozone, thus will only judge based on best expertise. This reviewer recommends the acceptance of this manuscript if the following comments can be sufficiently addressed.
Response: We thank the reviewer's very positive comments of our study! We provided detailed responses below (reviewers' comments in plain font, our replies in blue). We really appreciate the reviewers' time.

Specific comments:
Introduction:
In the first paragraph, it is worth adding the mechanisms of observed increasing ozone concentrations in China. The reasons include not only increasing anthropogenic VOC emissions but also decreased ozone titration due to decreased NOx emissions especially in megacities where ozone production is usually NOx-saturated. It is worth reviewing relevant literature.
Response: We thank the reviewer's suggestion. We now add the following discussion at the end of first paragraph:
"The increasing trend of surface ozone may be partially explained by the decreased titration due to the decreased $NO_X$ emissions especially in megacities (Liu et al., 2020a, b; Tan et al., 2020; Li et al., 2022), or the decreasing $PM_{2.5}$ which scavenges the radical precursors of ozone (Li et al., 2019a, 2020)."

Reference:
Li, K., Jacob, D. J., Zhang, Q., Liao, H., Bates, K. H. and Shen, L.: Anthropogenic drivers of 2013–2017 trends in summer surface ozone in China, Proc. Natl. Acad. Sci., 116(2), 422–427, doi:10.1073/pnas.1812168116, 2019a.

Li, K., Jacob, D. J., Shen, L., Lu, X., De Smedt, I., and Liao, H.: Increases in surface ozone pollution in China from 2013 to 2019: anthropogenic and meteorological influences, Atmos. Chem. Phys., 20, 11423–11433, https://doi.org/10.5194/acp-20-11423-2020, 2020.

Li, X., Yuan, B., Parrish, D. D., Chen, D., Song, Y., Yang, S., Liu, Z. and Shao, M.: Long-term trend of ozone in southern China reveals future mitigation strategy for air pollution, , 269(November 2021), 2022.
Liu, Y. and Wang, T. (2020a). Worsening urban ozone pollution in China from 2013 to 2017 – Part 1: The complex and varying roles of meteorology. Atmospheric Chemistry and Physics, 20(11), 6305–6321. https://doi.org/10.5194/acp-20-6305-2020.

Liu, Y. and Wang, T. (2020b). Worsening urban ozone pollution in China from 2013 to 2017 – Part 2: The effects of emission changes and implications for multi-pollutant control. Atmospheric Chemistry and Physics, 20(11), 6323–6337. https://doi.org/10.5194/acp-20-6323-2020.

Line 48-54. Literature seems to find very large yield decrease effects for soybean compared to other crops. I wonder why the authors found relatively small impact as indicated by Line 22, which is one order of magnitude smaller than previous research.

Response: We thank the reviewer pointing this out. We went back to check our calculation, and found out that we misplaced the concentration-response function for the relative yields (RY) for soybean from Mills et al. (2007). For soybean, the RY should be:

$$RY = -0.0116 \times AOT40 + 1.02$$

While we misplaced "1.12" here (see Table 1). After updating our calculations, we estimated that the annual soybean crop yields loss (RYL) reaches 6.51%-9.92% from 2010 to 2017, and much higher in Northeast China, reaching 20% for 8-yr average (e.g., Tianjin, Beijing, and Hebei in Table S6 in the supporting material). We then estimated 1.09-1.84 million metric tons for the ozone-induced soybean yield losses. Avnery et al. (2011a) reported RYL of 21-25% for China, and Zhang et al. (2017) reported 23.4%~30.2% annual soybean yield losses in 2014 in Northeast China. Wang et al. (2022) reported 1.2-1.6 million metric tons per year for the soybean losses from 2014 to 2017 when the same AOT40 metric was used. We now updated all the numbers for the RYL, CPL and economic losses associated with the soybean, as well as all the figures and tables in the main paper and supporting. We genuinely appreciated the reviewer's efforts in finding the error for us.

Reference:
Avnery, S., Mauzerall, D. L., Liu, J. and Horowitz, L. W.: Global crop yield reductions due to surface ozone exposure: 1. Year 2000 crop production losses and economic damage, Atmos. Environ., 45(13), 2284–2296, doi:10.1016/j.atmosenv.2010.11.045, 2011a.

Mills, G., Buse, A., Gimeno, B., Bermejo, V., Holland, M., Emberson, L. and Pleijel, H.: A synthesis of AOT40-based response functions and critical levels of ozone for agricultural and horticultural crops, Atmos. Environ., 41(12), 2630–2643, doi:10.1016/j.atmosenv.2006.11.016, 2007.

Wang, Y., Wild, O., Ashworth, K., Chen, X., Wu, Q., Qi, Y. and Wang, Z.: Reductions in crop yields across China from elevated ozone, Environ. Pollut., 292(September 2021), doi:10.1016/j.envpol.2021.118218, 2022.

L57-60: statement of the key innovation of this study does seem as persuasive, since Line 54-57 indicates that a recent study evaluates effects of ozone on yields of 3 crops for 4 years. The authors do 4 more years of analyses with 1 additional crop (i.e. soybean). Are there new data used or improved model simulation or emission inventories adopted in this research? This novelty statement seems a bit weak. In addition, did previous research not at all examine spatial variations of ozone damages to crop yields? If there are any, they need to be included as literature review here.

Response: We thank the reviewer pointing this out. Our study is innovative in carrying out first long-term temporal and spatial variations of the crop yields loss due to surface ozone in China. Previous studies have been focus on a specific region of China, such as North China Plain (Zhang et al., 2017; Hu et al., 2020; Feng et al., 2020), or Yangtze River Delta only (Wang et al.,

2012). The specific period of 2010-2017 was chosen to cover the emission changes before and after the China's Air Pollution Prevention and Control Action Plan (APPCAP) established in 2013. Previous studies have been reporting the crop yield changes in one year (e.g., Lin et al., 2018; Yi et al., 2018; Feng et al., 2019a,b), or several years after the APPCAP (Zhao et al., 2019; Wang et al., 2022), and our study allows for a comparison for the effectiveness before and after implementation of the APPCAP.

To make the innovation of our study more obvious, we rewrite the sentences from L64-74: "To date, very few studies have investigated the long-term trends and spatial patterns of ozone impacts on crop production in China. Previous studies have been mainly focus on a specific region of China, such as NCP (Zhang et al., 2017; Hu et al., 2020; Feng et al., 2020), or Yangtze River Delta (Wang et al., 2012). In this study, we focus on the long-term ozone-exposure impact analysis from 2010 to 2017 in China to assess the yield losses of four major crops (wheat,s maize, rice, and soybean) and evaluate their associated economic losses. The specific period of 2010-2017 was chosen to cover the emission changes before and after the APPCAP established in 2013. Previous studies have been reporting the crop yield losses in one year (e.g., Lin et al., 2018; Yi et al., 2018; Feng et al., 2019a,b), or several years after the APPCAP (Zhao et al., 2019; Wang et al., 2022), and our study aims to present a comprehensive analysis of ozone-induce crop yield losses and economic impacts in the agriculture sector before and after the China APPCAP. Such an analysis is expected to provide scientific support to policymakers for their decision making."

Reference:
Feng, Z., Kobayashi, K., Li, P., Xu, Y., Tang, H., Guo, A., Paoletti, E. and Calatayud, V.: Impacts of current ozone pollution on wheat yield in China as estimated with observed ozone, meteorology and day of flowering, Atmos. Environ., 217(March), doi:10.1016/j.atmosenv.2019.116945, 2019a.

Feng, Z., De Marco, A., Anav, A., Gualtieri, M., Sicard, P., Tian, H., Fornasier, F., Tao, F., Guo, A. and Paoletti, E.: Economic losses due to ozone impacts on human health, forest productivity and crop yield across China, Environ. Int., 131(June), doi:10.1016/j.envint.2019.104966, 2019b.

Feng, Z., Hu, T., Tai, A. P. K. and Calatayud, V.: Yield and economic losses in maize caused by ambient ozone in the North China Plain (2014–2017), Sci. Total Environ., 722, 137958, doi:10.1016/j.scitotenv.2020.137958, 2020.

Hu, T., Liu, S., Xu, Y., Feng, Z. and Calatayud, V.: Assessment of O3-induced yield and economic losses for wheat in the North China Plain from 2014 to 2017, China, Environ. Pollut., 258, 113828, doi:10.1016/j.envpol.2019.113828, 2020.

Lin, Y., Jiang, F., Zhao, J., Zhu, G., He, X., Ma, X., Li, S., Sabel, C. E. and Wang, H.: Impacts of O3 on premature mortality and crop yield loss across China, Atmos. Environ., 194(July), 41–47, doi:10.1016/j.atmosenv.2018.09.024, 2018.

Wang, X., Zhang, Q., Zheng, F., Zheng, Q., Yao, F., Chen, Z., Zhang, W., Hou, P., Feng, Z., Song, W., Feng, Z. and Lu, F.: Effects of elevated O3 concentration on winter wheat and rice yields in the Yangtze River Delta, China, Environ. Pollut., 171, 118–125, doi:10.1016/j.envpol.2012.07.028, 2012.

Wang, Y., Wild, O., Ashworth, K., Chen, X., Wu, Q., Qi, Y. and Wang, Z.: Reductions in crop yields across China from elevated ozone, Environ. Pollut., 292(September 2021), doi:10.1016/j.envpol.2021.118218, 2022.

Yi, F., McCarl, B. A., Zhou, X. and Jiang, F.: Damages of surface ozone: Evidence from agricultural sector in China, Environ. Res. Lett., 13(3), doi:10.1088/1748-9326/aaa6d9, 2018.

Zhang, W., Feng, Z., Wang, X., Liu, X. and Hu, E.: Quantification of ozone exposure- and stomatal uptake-yield response relationships for soybean in Northeast China, Sci. Total Environ., 599–600, 710–720, doi:10.1016/j.scitotenv.2017.04.231, 2017.

Zhao, H., Zheng, Y., Zhang, Y. and Li, T.: Evaluating the effects of surface $O_3$ on three main food crops across China during 2015–2018, Environ. Pollut., 258, 113794, doi:10.1016/j.envpol.2019.113794, 2020.

It is probably also useful to mention the uncertain impacts of climate change on crop yields and increasing future food demand associated with increased population and increased meat demand thus animal feed crops, in the introduction or somewhere in discussion. This will make the evaluation of ozone yield effects and potential mitigation appear to be more urgently relevant to air quality and food security.

Response: We thank the reviewer's insight suggestion. We now rewrite our Discussion, and add the influences of future climate change on crop yields, as well as the different population projections under the SSPs.

Line 258-262:

"Therefore, reducing surface ozone pollution could not only bring the benefits of reducing ozone-related premature deaths, but also bring the benefits of control the global hunger and malnutrition issues, thus helping to reach the Sustainable Development Goal 2 of "Zero Hunger". Meanwhile, Chinese population are projected to continue to increase and peak around 2025 under all the shared socioeconomic pathways (SSPs, Chen et al., 2020), making it more urgent to improve the crop productions by all means."

Line 269-272:

"In our study, we also did not consider the possible climate changes on the crop productions. However, previous studies have demonstrated that temperature increases could significantly reduce the crop productions as well (Asseng et al., 2015; Wiebe et al., 2015; Liu et al., 2016; Zhao et al., 2016, 2017)."

**Methods:**
Line 83: Model's underestimation of AOT40 seems a bit severe. Is there a way to constrain model results with observations? Does the under-estimation indicate underestimate of ozone concentrations? If this is a modeling issue pointed out before, relevant literature needs to be described? Possible mechanisms need to be addressed in Discussion.

Response: We thank the reviewer's suggestion. Uncertainties in meteorology, emissions, and chemical mechanisms, along with the spatial resolution of chemical transport models, can lead to biases in simulated ozone concentration. These biases are accumulated in concentration metrics, particularly for the threshold-based AOT40 metric. AOT40 metric is accumulated threshold-based metric and so the relationship between ozone concentration and AOT40 is nonlinear (Van Dingenen et al., 2018; Wang et al., 2020), and thus can not be biased corrected using commonly kirgging or Inverse distance weighted (IDW) interpolation methods. Van

Dingenen et al. (2009, 2018) concluded that when averaged at the regional scale, the global transport model simulated crop metrics obtained from the grid boxes reproduces the observations within their standard deviatioins. So considering both reviewer2 and reviewer3' comments, we removed the evaluation for the AOT40 between model and the observation, instead we showed the evaluation for the annual average maximum daily 8-hour average. We revised the sentences from line 77 to line 80:

"We first evaluated the model's performance by comparing the model simulated annual average maximum daily 8-hr average (MDA8) $O_3$ with the surface observation from 2013 to 2017, which were downloaded from National Environmental Monitoring Center (CNEMC) Network (http://106.37.208.233:20035/). It collects at least 100 million environmental monitoring data from 1497 established air quality monitoring stations annually for national environmental quality assessment. The ozone observation data before 2013 were not available (Lu et al., 2018, 2020). In general, our model captures spatial patterns of the ozone distribution in China (Fig. S6 in Zhang et al., 2021), but overestimates the annual MDA8 $O_3$ concentration, with mean bias of 5.7 ppbv and normalized mean bias of 13.7% for 5-yr average from 2013 to 2017 (Table 1 in Zhang et al., 2021)."

**Results:**
1 title 'ozone concentration change' is not precise – it is metric (AOT) value change. Consider revising the title.
Response: We now revise the title to "Temporal and spatial distribution of accumulated ozone change"

Line 141-144 seems to address my previous comment on Introduction but this review of literature has been put in a weird place.
Response: We now move the discussion abouts the ozone increase after year 2014 to the introduction:
"At the same time, however, anthropogenic emission of VOC increased by 11% due to the lack of effective emission controls (Zheng et al., 2018), and surface observations show that the ozone concentration in China still reveals a tendency of increasing (Wang et al., 2020; Li et al., 2018 & 2019a; Lu et al., 2018, 2020). The increasing trend of surface ozone may be partially explained by the decreased titration due to the decreased $NO_X$ emissions especially in megacities (Liu and Wang, 2020a, b; Li et al., 2022), or the decreasing $PM_{2.5}$ which scavenges the radical precursors of ozone (Li et al., 2019a, 2020), though this chemical pathway still exist debates (Tan et al., 2020)."

Reference:
Li, K., Jacob, D. J., Zhang, Q., Liao, H., Bates, K. H. and Shen, L.: Anthropogenic drivers of 2013–2017 trends in summer surface ozone in China, Proc. Natl. Acad. Sci., 116(2), 422–427, doi:10.1073/pnas.1812168116, 2019a.

Li, K., Jacob, D. J., Shen, L., Lu, X., De Smedt, I., and Liao, H.: Increases in surface ozone pollution in China from 2013 to 2019: anthropogenic and meteorological influences, Atmos. Chem. Phys., 20, 11423–11433, https://doi.org/10.5194/acp-20-11423-2020, 2020.

Li, X., Yuan, B., Parrish, D. D., Chen, D., Song, Y., Yang, S., Liu, Z. and Shao, M.: Long-term trend of ozone in southern China reveals future mitigation strategy for air pollution, 269 (118869), 2022.

Liu, Y. and Wang, T.: Worsening urban ozone pollution in China from 2013 to 2017 – Part 1: The complex and varying roles of meteorology. Atmospheric Chemistry and Physics, 20(11), 6305–6321. https://doi.org/10.5194/acp-20-6305-2020, 2020a.

Liu, Y. and Wang, T.: Worsening urban ozone pollution in China from 2013 to 2017 – Part 2: The effects of emission changes and implications for multi-pollutant control. Atmospheric Chemistry and Physics, 20(11), 6323–6337. https://doi.org/10.5194/acp-20-6323-2020, 2020b.

Lu, X., Hong, J., Zhang, L., Cooper, O. R., Schultz, M. G., Xu, X., Wang, T., Gao, M., Zhao, Y. and Zhang, Y.: Severe Surface Ozone Pollution in China: A Global Perspective, Environ. Sci. Technol. Lett., 5(8), 487–494, doi:10.1021/acs.esstlett.8b00366, 2018.

Lu, X., Zhang, L., Wang, X., Gao, M., Li, K., Zhang, Y., Yue, X. and Zhang, Y.: Rapid Increases in Warm-Season Surface Ozone and Resulting Health Impact in China Since 2013, Environ. Sci. Technol. Lett., doi:10.1021/acs.estlett.0c00171, 2020.

Tan, Z., Hofzumahaus, A., Lu, K., Brown, S. S., Holland, F., Huey, L. G., Kiendler-Scharr, A., Li, X., Liu, X., Ma, N., Min, K. E., Rohrer, F., Shao, M., Wahner, A., Wang, Y., Wiedensohler, A., Wu, Y., Wu, Z., Zeng, L., Zhang, Y., and Fuchs, H.: No Evidence for a Significant Impact of Heterogeneous Chemistry on Radical Concentrations in the North China Plain in Summer 2014, Environ. Sci. Technol., 54, 5973–5979, https://doi.org/10.1021/acs.est.0c00525, 2020.

Wang, Y., Gao, W., Wang, S., Song, T., Gong, Z., Ji, D., Wang, L., Liu, Z., Tang, G., Huo, Y., Tian, S., Li, J., Li, M., Yang, Y., Chu, B., Petäjä, T., Kerminen, V. M., He, H., Hao, J., Kulmala, M., Wang, Y. and Zhang, Y.: Contrasting trends of PM$_{2.5}$ and surface-ozone concentrations in China from 2013 to 2017, Natl. Sci. Rev., 7(8), 1331–1339, doi:10.1093/nsr/nwaa032, 2020.

Zheng, B., Tong, D., Li, M., Liu, F., Hong, C., Geng, G., Li, H., Li, X., Peng, L., Qi, J., Yan, L., Zhang, Y., Zhao, H., Zheng, Y., He, K. and Zhang, Q.: Trends in China's anthropogenic emissions since 2010 as the consequence of clean air actions, Atmos. Chem. Phys., 18(19), 14095–14111, doi:10.5194/acp-18-14095-2018, 2018.

Line 145-146: To explain the peak of AOT40 in one specific year, one needs to figure out whether the seasonality of ozone concentrations have changed over time since the growing season likely remain the same across years, correct?

Response: We agree with the reviewer that the growing season for the different crops will be unchanged, at least in our study from 2010 to 2017. The seasonality of ozone concentrations may change though. Meanwhile, for different crops, the growing season will differ too (see Table 1), which makes the direct comparisons very difficult. So in section 3.1, we showed the annual AOT40 changes, instead of seasonal AOT40.

Section 3.2 and 3.3 list many detailed results. I wonder if at the beginning of each paragraph the authors can summarize the findings in one topical sentence. What are the findings that should be noted without getting into all the details? The readers may get very lost with all the details.

Response: We thank the reviewer's suggestion. We now add one topical sentence in each from section 3.2 to 3.4:

Line 158:

"The accumulated AOT40 values vary among the four crops, mainly determined by the seasonality of ozone concentrations."

"From equation 3, we expect that the spatial distribution of CPL among the four crops would be different from their RYLs."

Line 217-218: results of this research is much much smaller than this previous research.
Response: See our response to the comments about Line 48-54.

**Discussion:**
It appears to me that Line 250-279 are still about results, although some comparisons with earlier research has been added.
Line 280-end appears to be actually like a real 'Discussion' that really expands the findings of the research. There are not very clear messages to policymaking regarding ozone control in which provinces should be prioritized. Consider improving the Discussion. More details could be provided regarding how to address ozone pollution in prioritized regions (i.e. high losses).
Response: We appreciate the reviewer's comment about the discussion. We feel these two questions are related, so we address them here together. We now rewrite the discussion. In the new Discussion section, we talked about the decreasing trend of ozone-induced crop yields losses in China after 2013, the future climate and population changes on crops, and also the uncertainties for our study originating from the model, the emission inventories and the concentration-response function we used. We also rewrote the Results and Summary section to show the results only.
**"4 Discussions**

Surface ozone emerged as an important environmental issue in China, and were shown increasing trend in major megacities for the past few years using both modelling and observation data (Lu et al., 2018, 2019; 2020; Li et als., 2020; Liu and Wang, 2020a,b; Ni et al., 2018; Wang et al., 2020), though strict clean air regulations have been implemented after 2013. Exposure to high concentrations of surface ozone not only poses threat to human health, but also cause damages to crop. Our study presented a comprehensive analysis on the impact of surface ozone exposure on four major crop production loss in China, including wheat, rice (double early and late rice, single rice), maize (north maize and south maize), and soybean. Unlike the surface ozone trend, we showed that the national crop yields for major crops in China usually peaks in 2014 or 2015, shortly after the strict clean air regulations after 2013. The decreasing trend of crop yield losses associated with surface ozone exposure was mainly explained by the fact that the surface ozone in China were increasing in urban areas, while decreasing in the rural areas (Li et al., 2022), where the major crops are planted. Nonetheless, the relatively higher ozone, especially compared with developed countries, such as United States and Japan (Lu et al., 2018), are still posing great threats to crop productions in China. Combing the annual crop production from the Statistical Yearbook of China, we estimated that the surface ozone in China could cause an average of 26.42 million metric tons losses (Mt) of wheat production from 2010 to 2017. These losses are even comparable to the annual average wheat production during the same period in Paris, which is the fifth largest wheat production in the world (http://www.fao.org/faostat/en/#data/QC, accessed December 12, 2021). We also estimated that the surface ozone exposure could cause 18.58 Mt losses of rice production in China, comparable to the annual rice production in Philippines, the world's 8[th] largest rice production. Transferring to economic values, we estimated the surface ozone exposure could cost more than 20 billion $ losses, representing more than 0.20% of annual average Gross Domestic Product (GDP) in

China from 2010 to 2017. The latest edition of the State of Food Security and Nutrition in the World estimated that between 720 and 811 million people in the world faced hunger in 2020, with 161 million increasing compared with 2019, and nearly 2.37 billion people did not have access to adequate food, with no regions spared (FAO, 2021). Therefore, reducing surface ozone pollution could not only bring the benefits of reducing ozone-related premature deaths, but also bring the benefits of control the global hunger and malnutrition issues, thus helping to reach the Sustainable Development Goal 2 of "Zero Hunger". Meanwhile, Chinese population are projected to continue to increase and peak around 2025 under all the shared socioeconomic pathways (SSPs, Chen et al., 2020), making it more urgent to improve the crop productions by all means.

Uncertainties exist in the design of our study, including the coarse resolution of the global transport model we used, the regional emission inventories, as we as the concentration-response functions. From the model evaluation, we learnt that our model tends to overestimate the annual MDA8 $O_3$ concentration in China. However, through sensitivity experiences, Wang et al. (2022) showed that model biases in ozone were likely to have a relatively small impact on estimated production losses. The uncertainties from the changes in growing seasons, and the concentration-response functions tend to have larger effects. We propose that further studies, using high-resolution bias-corrected ozone concentration data and region-specific response functions, need to be carried out to quantify the negative effects of surface ozone on crops. In our study, we also did not consider the possible climate changes on the crop productions. However, previous studies have demonstrated that temperature increases could significantly reduce the crop productions as well (Asseng et al., 2015; Wiebe et al., 2015; Liu et al., 2016; Zhao et al., 2016, 2017). Despite these limitations and uncertainties, our study strives to estimate the long-term negative effects from surface ozone exposure in China before and after the clean air action in China. These estimations could provide the government and policy-makers useful references to be taken into account of the detrimental effects of ozone exposure on crop productions in China when making regional-specific ozone control policies."

Grammar issues need to be fixed, to name a few, line 20 'in 2017'; Line 73 'outside of China' instead of 'outside China'
Response: We appreciate the reviewer's comments. We changed the above issues following the reviewer's suggestions.
We spent quite an effort to improve our writing when preparing for the revisions from all the coauthors. We also seek external help from senior researcher Dr. Russell Harwood (russell.harwood@duke.edu) from Duke University for advice. We believe our writing has been greatly improved.

**Tables and Figures:**
Table 1 seems to be methods and from previous research, instead of actual research design or results.
Response: we agree with the reviewer that the values from Table 1 are from previous research. We put it here to help the reviewers recognize the growing seasons and spatial distributions of major crops in Chinese provinces. Checking out previous studies (Lin et al., 2018; Wang et al., 2022), we prefer to keep Table 1 in the main paper. However, we changed the title to the following:

"Table 1: Overview of the concentration-response function for the relative yields (RY) for ozone exposure on different crops"

Figure 3 consider putting the names of corresponding crops next to the (a) (b) (c) (d)...
Response: Thanks for the suggestion. We now add the corresponding crops next to the (a) (b) (c) (d).

Figure 4 For some crops, the losses peak at 2014 while for others the losses peak at 2015.
Response: The reviewer is right that the yield losses for the different crops vary across years, which are caused by the different change patterns for seasonal ozone.

Figure 5 the caption needs to describe panels a) and b). Do you simply group the provinces based on the magnitude of values?
Response: Fig. 5 a) & b) shows the wheat production loss by magnitude for all the province in China. As also pointed out by Reviewer 2, we agree that there are too many bars for both Figs 5 & 6. So we revised these two plots to show only the top 5 provinces with the largest crop loss. The province-level results are kept in the Tables S7-S12 in the supporting material.

**Response to comments #3**

RC3 comments:

Li et al ("Surface ozone impacts on major crop production in China from 2010 to 2017") quantifies the crop production and economic loss from surface ozone ($O_3$) in China over several years. Overall, the method used is sound and has been used by many studies previously. However, significant improvements must be made to the description of results, discussion and implications for this to be a meaningful scientific paper worth of publication in ACP.

Response: We thank the reviewer's very positive comments of our study. We have revised the paper to take those comments into account. We provide detailed responses below (reviewers' comments in plain font, our replies in blue), and very much appreciate the reviewers' time.

Sections 3.2-3.4 should be simplified and reorganized (together or separately) to better highlight the main results, rather than list many values that can be found in tables and figures. Increase comparing/contrasting of different crops and regions and tie these to an improved discussion section.

Rsesponse: Response: We thank the reviewer's suggestion. We now add one topical sentence in each from section 3.2 to 3.4:

Line 158:

"The accumulated AOT40 values vary among the four crops, mainly determined by the seasonality of ozone concentrations."

Line 186:

"From equation 3, we expect that the spatial distribution of CPL among the four crops would be different from their RYLs."

The current discussion section is largely a restating of the intro, methods and results. Instead, expand the final paragraph to speak more about the implications of the work. Include discussion of the seasonal cycle of $O_3$ that is carried through to the cropping season differences. Add more about the chemistry and policies throughout China that causes the results. For example, why O3 increases when PM regulations were successful. This section should also include discussion of the uncertainties in the model $O_3$ concentration, AOT40 metric and economic valuation.

Response: We appreciate the reviewer's comment about the discussion. We now rewrite the discussion. In the new Discussion section, we talked about the decreasing trend of ozone-induced crop yields losses in China after 2013, the future climate and population changes on crops, and also the uncertainties for our study originating from the model, the emission inventories and the concentration-response function we used. We also rewrote the Results and Summary section to show the results only.

**"4 Discussions**

Surface ozone emerged as an important environmental issue in China, and were shown increasing trend in major megacities for the past few years using both modelling and observation data (Lu et al., 2018, 2019; 2020; Li et als., 2020; Liu and Wang, 2020a,b; Ni et al., 2018; Wang et al., 2020), though strict clean air regulations have been implemented after 2013. Exposure to high concentrations of surface ozone not only poses threat to human health, but also cause damages to crop. Our study presented a comprehensive analysis on the impact of surface ozone exposure on four major crop production loss in China, including wheat, rice (double early and late rice, single rice), maize (north maize and south maize), and soybean. Unlike the surface

ozone trend, we showed that the national crop yields for major crops in China usually peaks in 2014 or 2015, shortly after the strict clean air regulations after 2013. The decreasing trend of crop yield losses associated with surface ozone exposure was mainly explained by the fact that the surface ozone in China were increasing in urban areas, while decreasing in the rural areas (Li et al., 2022), where the major crops are planted.  Nonetheless, the relatively higher ozone, especially compared with developed countries, such as United States and Japan (Lu et al., 2018), are still posing great threats to crop productions in China. Combing the annual crop production from the Statistical Yearbook of China, we estimated that the surface ozone in China could cause an average of 26.42 million metric tons losses (Mt) of wheat production from 2010 to 2017. These losses are even comparable to the annual average wheat production during the same period in Paris, which is the fifth largest wheat production in the world (http://www.fao.org/faostat/en/#data/QC, accessed December 12, 2021). We also estimated that the surface ozone exposure could cause 18.58 Mt losses of rice production in China, comparable to the annual rice production in Philippines, the world's 8[th] largest rice production. Transferring to economic values, we estimated the surface ozone exposure could cost more than 20 billion $ losses, representing more than 0.20% of annual average Gross Domestic Product (GDP) in China from 2010 to 2017. The latest edition of the State of Food Security and Nutrition in the World estimated that between 720 and 811 million people in the world faced hunger in 2020, with 161 million increasing compared with 2019, and nearly 2.37 billion people did not have access to adequate food, with no regions spared (FAO, 2021). Therefore, reducing surface ozone pollution could not only bring the benefits of reducing ozone-related premature deaths, but also bring the benefits of control the global hunger and malnutrition issues, thus helping to reach the Sustainable Development Goal 2 of "Zero Hunger". Meanwhile, Chinese population are projected to continue to increase and peak around 2025 under all the shared socioeconomic pathways (SSPs, Chen et al., 2020), making it more urgent to improve the crop productions by all means.

Uncertainties exist in the design of our study, including the coarse resolution of the global transport model we used, the regional emission inventories, as we as the concentration-response functions. From the model evaluation, we learnt that our model tends to overestimate the annual MDA8 $O_3$ concentration in China. However, through sensitivity experiences, Wang et al. (2022) showed that model biases in ozone were likely to have a relatively small impact on estimated production losses. The uncertainties from the changes in growing seasons, and the concentration-response functions tend to have larger effects. We propose that further studies, using high-resolution bias-corrected ozone concentration data and region-specific response functions, need to be carried out to quantify the negative effects of surface ozone on crops. In our study, we also did not consider the possible climate changes on the crop productions. However, previous studies have demonstrated that temperature increases could significantly reduce the crop productions as well (Asseng et al., 2015; Wiebe et al., 2015; Liu et al., 2016; Zhao et al., 2016, 2017). Despite these limitations and uncertainties, our study strives to estimate the long-term negative effects from surface ozone exposure in China before and after the clean air action in China. These estimations could provide the government and policy-makers useful references to be taken into account of the detrimental effects of ozone exposure on crop productions in China when making regional-specific ozone control policies."

**More specific comments/suggestions are listed below:**
Line 74: Add at least the direction of adjustment. Increased due to vertical gradient near surface?

Response: Thanks for the comments. We now rephrase this sentence in line 82-85:
"We then adjusted the model simulated surface ozone from lowest grid box height (usually above 30 meters) to the crop height (usually 1 meter at the ambient observation sites), which usually reduce the simulated ozone concentrations by 30-50% (Van Dingenen et al., 2009; Zhang et al., 2012)."

Line 77: Why compare model AOT40 and not model concentrations? AOT40 has also not yet been introduced.
Line 80: A figure showing the observed-model concentration comparison would be helpful, especially the expected seasonal cycle, despite the bias. Do the patterns match?
Response: We thank the reviewer pointing this out. These are really good questionss! Since they are related, we put our responses together here.
We evaluated the modelled ozone concentration from 2013 to 2017 in our earlier paper using the same set of simulations, which was just published in the same journal (section 3.1 in Zhang Y. et al., 2021: *Impacts of emission changes in China from 2010 to 2017 on domestic and intercontinental air quality and health effect*). In Zhang et al., 2021, we evaluated the model's performance in simulating annual average maximum daily 8 h average (MDA8) $O_3$ by comparing with hourly surface observations retrieved from the China National Environmental Monitoring Center (CNEMC) network (http://106.37.208.233:20035/) from 2013 to 2017, since the data prior to 2013 are not available. From the comparison, we concluded that our model overestimates the annual MDA8 ozone in China, with mean bias of 5.7 ppbv and normalized mean bias of 13.7% for 5-yr average. However, the AOT40 comparison showed that our model simulated AOT40 values, after adjusting the model simulated ozone concentration at crop height, were lower than the observation. Reducing the sampling height from the lowest grid box center (~30m) to 1 crop height (1-3 m) on average decreases the AOT40 by half (Van Dingenen et al., 2009). To make it clear, we now rewrite this paragraph to show the model evaluations of the annual MDA8 $O_3$ from line 86 to line 93:

"We first evaluated the model's performance by comparing the model simulated annual average maximum daily 8-hr average (MDA8) $O_3$ with the surface observation from 2013 to 2017, which were downloaded from National Environmental Monitoring Center (CNEMC) Network (http://106.37.208.233:20035/). It collects at least 100 million environmental monitoring data from 1497 established air quality monitoring stations annually for national environmental quality assessment. The ozone observation data before 2013 were not available (Lu et al., 2018, 2020). In general, our model captures spatial patterns of the ozone distribution in China (Fig. S6 in Zhang et al., 2021), but overestimates the annual MDA8 $O_3$ concentration, with mean bias of 5.7 ppbv and normalized mean bias of 13.7% for 5-yr average from 2013 to 2017 (Table 1 in Zhang et al., 2021)."

Reference:
Van Dingenen, R., Dentener, F. J., Raes, F., Krol, M. C., Emberson, L. and Cofala, J.: The global impact of ozone on agricultural crop yields under current and future air quality legislation, Atmos. Environ., 43(3), 604–618, doi:10.1016/j.atmosenv.2008.10.033, 2009.

Zhang, Y., Shindell, D., Seltzer, K., Shen, L., Lamarque, J.-F., Zhang, Q., Zheng, B., Xing, J., Jiang, Z., and Zhang, L.: Impacts of emission changes in China from 2010 to 2017 on domestic

and intercontinental air quality and health effect, Atmos. Chem. Phys., 21, 16051–16065, https://doi.org/10.5194/acp-21-16051-2021, 2021.

Line 85: Should "matrixes" be "metrics"?
Response: We now change to "metrics"

Line 87: Why not use other metrics such as M12/M7 or others instead of or in addition AOT40?
Response: AOT40 metric is the European standard for the protection of vegetation, and widely used in both America and Asia (Tang et al., 2013; Lefohn et al., 2018; Lin et al., 2018). The AOT40 metric is also considered as more accurate at high levels of ozone concentration (Tuovinen, 2000; Hollaway et al., 2012), which is the case for ozone pollution in China (Lu et al., 2018, 2020). To clarify this, we modify the sentence in line 97-99:

"In this study, we adopted the ozone metric of AOT40 which is the European standard for the protection of vegetation, and also the commonly used and reliable indicator in both America and Asia for crop yield assessment (UNECE, 2017; Tang et al., 2013; Lefohn et al., 2018; Lin et al., 2018; Feng et al., 2019a,b). The AOT40 metric is also considered as more accurate at high levels of ozone concentration (Tuovinen, 2000; Hollaway et al., 2012), which is the case for China (Lu et al., 2018, 2020)."

Line 126: Is this the global price from FAOSTAT?
Response: We thank the reviewer's question. The purchase price in each country is considered as market price according to FAOSTAT (FAOSTAT, 2020; Feng et al., 2019a), and the price in line 126 is the price in China. To avoid confusion, we changed "purchase price" to "market price" through the paper, and also modify the sentence 138-140 to make it clearly:
"where $Crop\ Price_p$ stands for the annually markets price in China for each crop with unit of USD/Mt. Crops markets prices were acquired from the FAOSTAT (http://www.fao.org/faostat/, last accessed 26th, March, 2s020; Feng et al,, 2019a)."

Reference:
FAOsSTAT, 2020, http://www.fao.org/faostat/, last accessed 26th, March, 2020

Feng, Z., De Marco, A., Anav, A., Gualtieri, M., Sicard, P., Tian, H., Fornasier, F., Tao, F., Guo, A. and Paoletti, E.: Economic losses due to ozone impacts on human health, forest productivity and crop yield across China, Environ. Int., 131(February), 104966, doi:10.1016/j.envint.2019.104966, 2019a.

Lines 134-135: This section uses the annual values to show the general trends and distribution, not because of the varying growing seasons.
Response: We thank the reviewer pointing this out. We now rewrite the sentence in line 147-149:
"Since the surface ozone in China has a distinct seasonal variation, thus making the direct comparison of the accumulated AOT40 values between the four crops impossible (Table 1), here we present the temporal and spatial distribution of annual accumulated AOT40 in China from 2010 to 2017."

Line 185: "later" than?
Response: We changed to "later than 2014". Thanks for pointing out.

Line 192: This is actually due to the seasonal cycling / varying $O_3$ between the growing seasons, not the difference in calculation of the growing season itself

Response: We agree with the reviewer and revised the sentence in line 200 below:

"The CPL for double early and late rice both peak in 2014, but with different years for the lowest values (Tables S9 and S10), highlighting the seasonal variations of $O_3$ concentration between different growing seasons (Table 1)."

Figure 5: What is the a) and b) each referring to? Missing from the caption.
Figures 5-6: There are too many bars, with the variation between crops in many provinces roughly the same. Consider simplifying to highlight main points.
Response: We appreciate the reviewer's comments about the Figures 5 & 6. Since these two questions are related, we address them together.
Fig. 5 a) & b) shows the wheat production loss by magnitude for all the province in China. We agree with the reviewer that both the Figs 5-6 have too many bars (as also pointed out by reviewer 2), so we revised these two plots to show only the top 5 provinces with the largest crop loss. The province-level results are kept in the Tables S7-S12 in the supporting material.

[Figure]

**Figure 5: Annual wheat production loss by province from 2010 to 2017 (1000 metric tons) due to surface ozone exposure.**

[Figure]

[Figure]

**Figure 6: The production losses for rice, including double early rice (a), double late rice (b), and single rice (c) in all the China provinces. Units of thousands metric tons.**